# A Mixed-Methods Evaluation of a Post-COVID-Condition Rehabilitation and Recovery Intervention Delivered in a Football Club Community Trust

**DOI:** 10.3390/ijerph22111672

**Published:** 2025-11-04

**Authors:** Steven Rimmer, Adam J. Herbert, Adam L. Kelly, Irfan Khawaja, Sam Lee, Lewis A. Gough

**Affiliations:** 1Human Performance and Health Laboratory, Birmingham City University, Birmingham B15 3TN, UK; steven.rimmer@mail.bcu.ac.uk (S.R.); adam.herbert@bcu.ac.uk (A.J.H.); adam.kelly@bcu.ac.uk (A.L.K.); irfan.khawaja@bcu.ac.uk (I.K.); 2Burton Albion Community Trust, Burton-Upon-Trent DE13 0AR, UK

**Keywords:** Post-COVID-Condition, exercise therapy, treatment outcome, football club community trust, quality of life, physical function, mixed methods

## Abstract

**Aim:** Post-COVID condition (PCC) is largely considered the biggest public health emergency in recent times. The role of exercise therapy in PCC is currently unknown, and evaluative studies are currently lacking in this area. This study therefore aimed to evaluate the effects of a football club community trust exercise rehabilitation programme on physical function and quality of life in individuals with PCC. **Method:** A mixed-methods retrospective design was employed, utilising a framework to assess the programme’s reach, effectiveness, adoption, implementation, and maintenance (RE-AIM). Quantitative data (questionnaires and physiological assessments) were collected at baseline, 6 weeks, and 12 weeks during the programme, and at 6 months post-intervention (*n* = 7). Qualitative data were gathered through semi-structured focus groups at week 12 (*n* = 7) and 12 months (*n* = 5) post-intervention. Quantitative data and qualitative data were analysed using repeated measures ANOVAs and thematic analysis, respectively. **Results:** The programme led to significant improvements in physical function, including increased six-minute walking distance (6MWT, *p* < 0.001), one-minute sit-to-stand repetitions (1MSST, *p* < 0.035), and lung function (spirometry; MIP: *p* = 0.048, MEP: *p* = 0.024). Participants also reported enhanced QoL (HRQoL-14, *p* = 0.004), reduced anxiety (GAD-7, *p* = 0.008) and depression (PHQ-9, *p* = 0.008), and increased confidence and self-efficacy. The programme was well-received, with participants valuing the supportive environment and personalised approach. **Conclusions:** Football community trust exercise rehabilitation programmes can effectively improve physical function and quality of life in individuals with PCC, offering a promising model for community-based rehabilitation. Further studies are needed with larger sample sizes to assess the scalability of similar programmes.

## 1. Introduction

The emergence of SARS-CoV-2 in 2019 significantly impacted global health, with long-term effects like post-COVID condition (PCC) affecting millions worldwide [1,2,3]. PCC, also known as long COVID, presents a wide range of symptoms affecting multiple organ systems, making it a complex and challenging condition to manage. The National Institute for Health and Care Excellence (NICE) defines PCC as signs and symptoms that persist for more than 12 weeks after a SARS-CoV-2 infection, cannot be explained by another diagnosis, and often occur in clusters that fluctuate over time and affect various body systems [1]. This definition highlights the persistent and multifaceted nature of PCC, emphasising that it is not simply a prolonged version of the acute SARS-CoV-2 illness but a distinct condition with its unique challenges. Recent estimates suggest approximately 200 million people worldwide are experiencing PCC [2], with 1.9 million cases in the United Kingdom (UK) alone [3]. Due to limited treatment options and the sheer number of individuals affected, PCC poses a potential crisis for the UK healthcare system, placing a significant burden on resources and necessitating innovative approaches to management and rehabilitation.

Contrary to initial assumptions that focussed primarily on respiratory symptoms, SARS-CoV-2 infection extends beyond the respiratory system, manifesting in over 200 symptoms across multiple organ systems [4,5]. The virus utilises the widespread angiotensin-converting enzyme 2 (ACE2) receptors for cellular entry, facilitating replication and triggering an inflammatory response characterised by immune cell activation and cytokine release [6]. This dysregulated immune response, often termed a “cytokine storm”, can inflict substantial damage on various organs, including the heart, lungs, kidneys, brain, and liver [7]. While research on the long-term sequelae is ongoing, accumulating evidence suggests potential for SARS-CoV-2 to induce lasting damage in these organs [4]. This damage can manifest in a range of chronic conditions, including but not limited to heart failure, arrhythmias, respiratory dysfunction, renal impairment, cognitive deficits, and neurological complications. The severity of these outcomes likely varies depending on individual factors such as initial infection severity, immune response, and pre-existing health conditions [4] These findings underscore the importance of understanding the complex pathophysiology of PCC and tailoring treatment approaches accordingly.

To address the multifaceted nature of PCC, researchers and clinicians have explored various rehabilitation strategies, with a particular focus on exercise-based interventions. To date, research on exercise-based interventions has shown promise, with exercise-based programmes improving physical function, quality of life (QoL), and mental well-being [8,9,10,11,12]. A systematic review and meta-analysis of 23 studies including 1579 individuals displayed positive effects on PCC-related symptoms including fatigue, dyspnoea, and depression, as well as improvements in overall QoL [11]. These interventions have included aerobic and resistance training, inspiratory muscle training, and digital/community-based programmes, highlighting the diverse approaches being explored to address the varied symptoms of PCC. Zheng et al. [11] further supports the benefits of exercise in PCC recovery, demonstrating its potential to improve exercise capacity, lung function, and psychological well-being. However, limitations exist in the current research, including short follow-up periods, lack of personalised approaches, and the need to understand the underlying mechanisms of improvement. These limitations underscore the importance of continued research to refine rehabilitation strategies and optimise outcomes for individuals with PCC.

Perhaps a more alarming oversight is the lack of measurement of post-exertional symptom exacerbation (PESE) in PCC [12]. PESE is characterised by a worsening of symptoms after physical or mental exertion, posing a significant challenge for individuals with PCC and potentially hindering their recovery. This phenomenon emphasises the importance of tailoring exercise programmes to individual needs, gradually increasing intensity and duration, and using objective measures to monitor progress and prevent overexertion. Existing research on exercise interventions for PCC is limited by several factors. Many studies have short follow-up periods, failing to capture the long-term impacts and potential for symptom relapse, particularly concerning PESE [11]. Additionally, there is a lack of personalised approaches that cater to the diverse range of PCC symptoms, and the underlying mechanisms of reduced cardiorespiratory fitness remain unclear [8]. This makes it difficult to develop targeted rehabilitation strategies that address the specific needs of each individual with PCC. These limitations underscore the need for caution in interpreting existing evidence and highlight the importance of further research with longer follow-up periods, personalised approaches, and a focus on understanding the underlying mechanisms of action.

Given the positive impact of football-led health initiatives [13,14], independent community-led interventions, such as those offered by football club community trusts (FCCTs), may provide a positive model for PCC rehabilitation. These initiatives leverage the popularity and reach of football to engage individuals in health-promoting activities, potentially offering a novel approach to addressing the widespread impact of PCC. However, research in this area is lacking, and the effectiveness of FCCT-led programmes for PCC rehabilitation remains unknown. If successful, the use of FCCTs could offer a cost-effective and accessible method for individuals to access rehabilitation. It may also reduce the burden on traditional primary care and reliance on more costly services (i.e., in hospital care and rehabilitation services). This study therefore aims to evaluate the effects of an FCCT-led exercise rehabilitation programme on physical function and QoL in individuals with PCC. It is hypothesised that such a programme, tailored to the specific needs of individuals with PCC and incorporating strategies to mitigate the risk of PESE, will lead to improvements in mental and physical health outcomes.

## 2. Materials and Methods

The RE-AIM framework is a model utilised to help plan and evaluate the public health impact of interventions [15]. This study used a mixed-methods retrospective design to employ the RE-AIM framework to evaluate the Reach, Effectiveness, Adoption, Implementation, and Maintenance (RE-AIM) of a Post-COVID-Condition Rehabilitation and Recovery Programme (PCCRRP) in an FCCT (Table 1).

### 2.1. Intervention Context and Setting

Established in 2010, Burton Albion Community Trust (BACT) is a recognised FCCT organisation delivering successful community-based health interventions [13]. In response to the local need for support among people living with post-COVID condition (PLWPCC), participants were referred to an FCCT PCCRRP designed to improve outcomes for service users diagnosed with PCC. The PCC clinic referred appropriately clinician-assessed service users into a 12-week community-led, exercise and psychosocial support intervention. Following the PCC clinic assessment, the need for rehabilitation was established when symptoms were complex in nature and caused significant functional impairment, thus requiring a supervised and structured multidisciplinary approach to recovery. Participants were then recruited from BACT by Birmingham City University (BCU) to participate in this study.

The PCCRRP pilot employed a 12-week personalised exercise referral scheme (ERS) delivered twice weekly in a community setting. PCCRRP plans were co-designed with participants and BACT and typically included supervised low to moderate intensity exercise sessions consisting of a combination of aerobic, stability and mobility, and strength-based exercises. PCCRRP plans were created and overseen by BACT personnel with level 7 qualifications in sport and exercise science and health and physical activity. Moreover, BACT personnel had experience managing patients with multiple health conditions and comorbidities. Extreme caution was applied prior to commencing any exercise session. Before initiating each gym-based exercise session, participants were screened for PEM and ME/CFS [16,17]. Any participant displaying signs of post-exertional malaise (PEM) and Myalgic Encephalomyelitis/Chronic Fatigue Syndrome (ME/CFS) would discontinue the PCCRRP and be referred to primary care. Additionally, resting heart rate (RHR), blood pressure (BP), and arterial oxygen concentration levels (SPO_2_) were measured and recorded prior to exercise. All gym-based exercise was conducted in-person in strictly controlled gym environments. The modified BORG scale (mBORG) [18] was utilised to measure the perceived exertion level of exercise throughout participation in the full exercise session. Participants could contact the referring primary care provider at any point during the 12 weeks. Throughout the intervention period, participation was free of charge. Participants received biweekly contact, both at gym-based sessions and additionally through virtual (Microsoft Teams) or phone contact for additional advice and guidance if required. For those who did not attend (DNA), three attempted calls were made within 48 h. If there was no response or the participant did not wish to be part of or continue the programme, the primary care provider referrer would be informed.

### 2.2. Participants and Procedures

All seven adults were adults aged 18 years or older and had undergone a medical assessment at the PCC clinic by their primary care provider (PCP) (Table 2). The PCP then referred the patient to BACT. Participants included in this project were identified by PCPs as requiring support with rehabilitation as part of PCC assessment pathways. Upon referral to BACT, prior to initial interviews, baseline questionnaires, and physiological assessments, service users were requested to provide consent for their data to be used for research purposes via a project participant information sheet and consent form. Additionally, they were invited to participate in focus groups (FGs). The lead author (SR) and co-authors (LG and IK) facilitated the FGs. Only the data of those who completed and returned consent forms were included in this study. All participants were provided with comprehensive details of the project, including its rationale, methodology, potential benefits, and potential risks, prior to providing informed consent. Furthermore, all participants understood that participation was voluntary and that there were no consequences for choosing not to participate and that they could discontinue participation if they wished to. Ethical approval for the study was obtained through BCU Health, Education and Life Sciences Faculty Academic Ethics Committee (ID#10203). All participants who enrolled into the study completed the study in full and a summary of procedures can be seen in Figure 1.

The inclusion criteria for the Post-COVID-Condition Rehabilitation and Recovery Program (PCCRRP) service users were a previous diagnosis of SARS-CoV-2 with a current negative test result, being 18 years of age or older, the ability to walk independently for at least 20 metres, and having access to transportation to the gym-based setting. Exclusion criteria included active SARS-CoV-2 symptoms (positive test), already receiving community-based rehabilitation, a diagnosis of Myalgic Encephalomyelitis/Chronic Fatigue Syndrome (ME/CFS), or a formal diagnosis of post-traumatic stress disorder (PTSD), clinically significant anxiety, or depression.

### 2.3. Clinical and Physiological Measures

Body mass (BM) was measured at baseline, 6 weeks, 12 weeks, and 6 months using Seca 813 digital flat scales (Seca, Birmingham, UK). Resting heart rate (RHR) and blood pressure (BP) were measured at the same time points using the Omron M3 Comfort blood pressure monitor (Omron, Milton Keynes, UK) [19]. Blood oxygen saturation (SpO2) was also measured at these intervals using an NHS-approved PX-100 EU Salter fingertip pulse oximeter (Salter, Manchester, UK) [20]. Breathing difficulty was assessed at baseline, 6 weeks, 12 weeks, and 6 months using the modified Borg scale (mBorg) [18].

Lung function, specifically maximal inspiratory pressure (MIP) and maximal expiratory pressure (MEP), was measured at baseline, 6 weeks, 12 weeks, and 6 months using the CareFusion MicroRPM respiratory pressure spirometer (CareFusion, San Diego, CA, USA) [21]. These measures were employed as a marker of improvement from the rehabilitation programme. No specific respiratory training was conducted using these devices. The DePaul Post-Exertional Malaise Questionnaire (DPEMQ) was administered before each exercise session to assess post-exertional malaise (PEM) and determine exercise suitability [16]. Physical activity levels were assessed at baseline, 6 weeks, 12 weeks, and 6 months using the Short Active Lives Questionnaire (SALS) [22].

To evaluate lower-body strength and exercise capacity, the one-minute sit-to-stand test (1MSST) and the six-minute walk test (6MWT) were conducted at baseline, 6 weeks, 12 weeks, and 6 months [23]. Perceived exertion during the 6MWT was measured using the mBorg scale [18]. Mental well-being was assessed at the same time points using the Generalised Anxiety Disorder 7 (GAD-7) [24,25], Patient Health Questionnaire 9 (PHQ-9) [26,27], and Short Warwick-Edinburgh Mental Well-being Scale (SWEMWBS) [28] to evaluate anxiety, depression, and mental well-being, respectively. Lastly, the CDC HRQOL-14 was used to evaluate health-related quality of life (HRQoL) at baseline, 6 weeks, 12 weeks, and 6 months [14].

### 2.4. Focus Groups

FG discussions were conducted at week 12 and 12 months post-intervention to gather insights from participants regarding their lived experiences with the PCCRRP and its impact (Appendix A). The RE-AIM framework served as the theoretical underpinning for the evaluation, guiding the FG discussions’ structure and content. The interview schedule for the FGs was crafted to align with the RE-AIM framework’s components. This approach ensured that the discussions focussed on the programme’s reach, effectiveness, adoption, implementation, and maintenance.

FGs have been widely employed in previous research investigating participant experiences in football-led health improvement interventions, effectively capturing both the implementation component of RE-AIM [15] and key characteristics such as the Premier League Men’s Health Evaluation and the Fit Red’s Men’s Health Evaluation [29,30]. These methods offer an inclusive, efficient, and convenient approach to gathering information on participant experiences and their engagement with interventions.

Each FG consisted of a sex-mixed composition. Seven participants attended the 12-week FG, while five attended the 12-month FG. The FGs were both held at Burton Albion Football Community Centre lasting approximately 90 min and were digitally recorded. Guided by the principle of reflexivity, the thematic analysis focussed on the trustworthiness and authenticity of findings [31]. The lead author (SR) and co-authors (LG and IK) served as note-takers, recording key themes and sub-themes emerging from the FG discussions. After each session, these themes and sub-themes were verbally presented back to the participants for confirmation and clarification, ensuring the accuracy and authenticity of the captured information. This participant-led verification process ensured the authenticity and accuracy of the acquired information. Thus, demonstrating participant experience and privilege of voice over the researchers’ own perspectives was a key part of the design of the research process. Such a deliberate strategy enhances the rigour and validity of the thematic analysis [31].

The lead researcher (SR) cultivated rapport with the participants throughout the 12-week intervention period, fostering a sense of familiarity and trust, which was vital for facilitating open and honest dialogue. This pre-existing connection was established during the baseline consultation and PCCRRP delivery, as well as during recruitment and participation in the FGs. Similarly, existing relationships among participants further enhanced engagement and interaction during the discussions. Additionally, it highlights the positive impact of these connections on participant engagement and interaction and consequently helps generate meaningful results.

### 2.5. Data Reduction and Analysis

Upon completion, all questionnaires were entered into a password-protected Excel spreadsheet. Coded identification numbers were used for participant anonymity. The data were then transferred to SPSS (v29 for Windows, IBM Corp., Chicago, IL, USA) for cleaning and analysis.

In accordance with Rutherford et al. [14] a “completers only” approach was adopted to analyse the outcome data. Completers were defined as individuals who provided usable questionnaire data at baseline, 6 weeks, 12 weeks, and 6 months. This approach aligns with that of Rutherford et al. [14] which provides the most effective method for determining changes in physical activity and HRQOL over time, considering the constraints of the data collection method.

All data were assessed for normality using the Shapiro–Wilk test and visually inspected using boxplots. Physiological and questionnaire data were analysed using repeated measures analysis of variance (ANOVA) over changes in time between baseline, 6 weeks, 12 weeks, and 6 months. For ANOVA interactions and main effects, the effect size is reported as partial eta squared (pη2). The assumption of normal distribution was violated for HRQoL and SALS, therefore the Friedman test, a non-parametric alternative, was conducted as the alternative test. Between changes in time, effect sizes (g) were calculated by dividing mean difference by the pooled standard deviation (SD) [32,33,34] and applying Hedge’s g bias correction to account for the small sample size [32,33,34]. These effect sizes were categorised as trivial (≤0.2), small (0.2–0.49), moderate (0.5–0.79), or large (≥0.8) [34]. Effect sizes for non-normally distributed data (r) were calculated from z/√n, with 0.10, 0.24, and 0.37 considered as small, medium, and large, respectively [32]. Statistical data are presented with 95% confidence intervals to indicate the precision of the estimated values. All data were analysed using SPSS (v29 for Windows, IBM Corp., Chicago, IL, USA) and statistical significance was determined as *p* < 0.05.

FGs were administered at week 12 and 12 months, respectively. Following the FGs, the recorded audio files underwent verbatim transcription by the lead author (SR) and co-author (SL). Guided by the principle of reflexivity [31], the thematic analysis was designed to elicit authenticity in its findings. Thematic analysis utilised the process of familiarisation, coding, generating themes, reviewing themes, defining and naming themes, and writing up the results [31]. For the interview data, after transcription and immersion in the transcripts, coding revealed intriguing features within the data. These features were subsequently grouped into coherent themes using the RE-AIM framework components. A visual map was hand-crafted to illustrate the themes and their interrelationships. Four researchers (LG, IK, AH, and AK) met to refine the specifics of the themes and to generate clear definitions and names for them, which were then shared with the lead author (SR). This approach has commonly been used in the investigation of football-led health improvement programmes [14]. A collaborative approach to data collection was utilised to mitigate any potential for researcher bias.

Following both quantitative and qualitative analysis, a cross-validation via triangulation was conducted to determine convergent or divergent narratives. The qualitative data was then seen as complementarity to the quantitative date to help answer *why* and *how* results occurred. This was seen as beneficial for this study as most studies investigating exercise and PCC had isolated methods (i.e., either quantitative or qualitative methods).

## 3. Results and Discussion

Results are presented detailing the quantitative and qualitative results separately. The discussion then integrates these findings, presenting interpretation in a combined manner, providing a comprehensive understanding of the research.

### 3.1. Demographics

A summary of demographics can be seen in Table 2 with the n number and percentage distribution reported.

### 3.2. Exercise Variables

Results showed a significant increase in the distance covered during the 6MW throughout the study (*p* < 0.001; Pη2 = 0.969). This improvement was significant and substantial at all time points except for between 12 weeks and 6 months. Between the baseline vs. week 6, a large effect size (ES = 3.86, 95% CI = 5.77, 1.95) was observed, with a significant increase (*p* < 0.001; CI = 516.7, 369) in walking distance by week 6. Significant improvements with large effect sizes (*p* = <0.001; CI = 723.9, 556.1) (ES =5.2, 955 CI = 7.56, 2.83), (*p* = <0.001; CI = 893.8, 532) (ES = 5.83, 95% CI = 8.42, 3.24) were observed at week 12 and 6 months compared to the baseline, respectively. While significant differences were observed between all time points, with the exception of week 12 vs. 6 months (*p* = 0.699), large effect sizes were present throughout; week 6 vs. week 12 (*p* < 0.001; CI = 280.5, 113.8) (ES = 2.62, 95% CI = 4.16, 1.08) and week 6 vs. 6 months (*p* = 0.002; CI = 415.3, 124.7) (ES =3.69, 95% CI = 5.55, 1.83).

The study displayed a significant increase in the number of 1MSS repetitions performed throughout the study (*p* < 0.035; Pη2 = 0.373). This improvement was most evident in the early stages. Between baseline vs. week 6, a significant increase (*p* = 0.010; CI = 14, 2.3) with a large effect size (ES = 1.51, 95% CI = 2.79, 0.23) was observed, suggesting a substantial increase in 1MSS repetitions by week 6. The difference between baseline vs. week 12, while significant (*p* = 0.003; CI = 20.1, 5.3), had only a moderate effect size (ES = 0.71, 95% CI = 1.88, 0.45) compared to baseline vs. week 6. No significant differences were found at later time points (*p* > 0.05). However, baseline vs. 6 months displayed a large effect size (ES = 1.45, 95% CI = 2.73, 0.18), suggesting a significant increase in 1MSS repetitions compared to baseline.

### 3.3. Physiological Variables

Results showed a significant increase in MIP scores over time (*p* = 0.048; Pη2 = 0.349). This increase was most evident between baseline and week 6 (*p* = 0.018; CI = 44.9, 4.8), with a moderate effect size (ES= 0.73, 95% CI = 1.09, 0.44). No significant differences were observed at later time points (*p* > 0.05). Results showed a significant improvement in MEP scores over time (*p* = 0.024; Pη2 = 0.463). While post hoc analysis did not detect significant differences between specific time points (*p* > 0.05), a moderate effect size was observed between baseline and week 6 (ES= 0.74, 95% CI = 1.91, 0.93). No significant differences were observed at later time points (*p* > 0.05). Results show decreased dyspnoea across the study (*p* = 0.341, Pη2 = 0.166). A significant decrease in dyspnoea was observed between baseline and week 6 (*p* = 0.019; CI = 0.334, 3.23) with a large effect size (ES = 1.14, 95% CI = 0.08, 2.36). No significant differences were found between later time points (*p* >0.05). No significant differences were observed in BM across the duration of the study (*p* = 0.149; Pη2 = 0.307). No significant differences were observed in RHR (*p* = 0.179; Pη2 = 0.249) or SPO_2_ (*p* = 0.273; Pη2 = 0.194) across the duration of the study. No significant differences were observed in systolic BP (*p* =0.503; Pη2 = 0.108) or diastolic BP (*p* =0.963; Pη2 = 0.006) across the duration of the study.

### 3.4. Questionnaire Variables

The results showed that a significant increase in SALS scores was observed across the study (χ2(2) = 6.231, *p* = 0.044). The difference between baseline vs. week 6, although not significant (*p* = 0.18), had a large effect size (ES = 0.82, 95% CI = 2, 0.86) which suggests an initial increase in activity by week 6. For baseline vs. week 12, a similar pattern emerged. No significant differences were observed (*p* = 0.249); however, its large effect size (ES = 0.87, 95% CI = 2.05, 0.32) points towards a substantial activity increase by week 12. No other significant differences were observed across later time points (*p* > 0.05).

Results show a significant impact on GAD-7 scores, indicating a reduction in anxiety (*p* = 0.008; Pη2 = 0.549). While post hoc analysis did not reveal significant differences between specific time points (*p* > 0.05), effect sizes displayed a large effect size (ES = 1.54, 95% CI: 0.25, 2.83) between baseline and week 6, indicating a substantial initial decrease in anxiety. No significant differences were observed at later time points (*p* > 0.05).

Results show a significant difference in PH-9 scores across the study (*p* = 0.008; Pη2 = 0.536). Despite no significant differences between specific time points upon further analysis (*p* > 0.05), large effect sizes were observed between baseline and week 6 (ES = 1.21, 95% CI = 0.02, 2.44) and baseline and week 12 (ES = 1.08, 95% CI = 0.13, 2.30), respectively, indicating substantial improvements in depressive symptoms early in the intervention. No significant differences were observed at later time points (*p* > 0.05).

Results show a significant increase in SWEMWBS scores, indicating improved mental well-being across the study (*p* = 0.001; Pη2 = 0.624). This improvement was evident at all time points compared to baseline. Significant increases with large effect sizes (*p* = 0.017; CI = 10, 1.45) (ES = 1.09, 95% CI = 2.3, 0.13) (*p* = 0.014; CI = 12.2, 2.1) (ES = 1.41, 95% CI = 2.67, 0.15) (*p* = 0.013; CI = 14.1, 2.5) (ES= 1.41, 95% CI = 2.67, 0.15) were observed between baseline and all follow-up points at weeks 6 and 12 and 6 months, respectively. While the difference between week 6 and 6 months was significant (*p* = 0.042; CI = 5.01, 0.133), the effect size between week 6 and 6 months (ES = 0.7, 95% CI = 1.82, 0.5) was moderate compared to earlier time points. No significant differences were observed at later time points (*p* > 0.05).

Results show a significant improvement in participants’ QoL across the study duration (*p* = 0.004; Pη2 = 0.702). This was supported by positive changes reported on the HRQoL-14 across various dimensions (Table 3), including self-care, work or recreation, pain, mood, sleep quality, and energy levels. Significant improvements with large effect sizes (*p* = 0.009; CI = 5.6, 32.1) (ES = 1.36, 95% CI: 0.10, 2.62) (*p* = 0.004; CI = 8.2, 32.3) (ES = 1.63, 95% CI = 0.32, 2.93) (*p* = 0.004; CI = 8.9, 34.2) (ES = 1.69, 0.37, 3) were observed between baseline and all follow-up points, weeks 6 and 12 and 6 months, respectively. No significant differences were found between later time points (*p* > 0.05).

### 3.5. Focus Group at Week 12

#### 3.5.1. Reach

Seven participants took part in the PCCRRP (*n* = 5 female, *n* = 2 male). Participants expressed being drawn to the FCCT setting due to it feeling less clinical than a traditional GP surgery. However, others stated their desperation to improve their health and well-being was such that they were willing to try anything to obtain the help and support they felt they needed.

#### 3.5.2. Effectiveness

The positive changes observed in QoL scores were supported by qualitative data from the FGs. The qualitative data presented in Table 4 and Table 5 suggests that participation in the PCCRRP had a positive impact on participants’ QoL. These findings are consistent with those reported previously in the literature [9,11,35,36]. A recurrent theme identified throughout the FG discussions was the concept of a “light at the end of the tunnel,” signifying a sense of hope, in addition to “feeling normal”, which related to regaining a sense of pre-illness functioning. There was a strong emphasis on the participants’ desire to return to their usual activities and capabilities and, in some instances, beyond their original capabilities pre-diagnosis, consistent with findings in an FCCT setting by Rutherford et al. [14] with people living with cancer. Participants provided detailed accounts in Table 6 of how the PCCRRP had improved their fitness, mental well-being, confidence and self-efficacy, and ability to return to pre-diagnosis levels of function, including work and exercise participation.

#### 3.5.3. Adoption

The themes relating to the profiles of participants who engaged in the PCCRRP including their physical activity, health status, and reported barriers and facilitators are presented in Table 6. Participants reported a significant factor in their improvement was the PCCRRP focusing on personalised exercise plans. Participants expressed that the tailored exercise plans empowered them to manage their symptoms and helped them to regain a sense of control over their lives. Furthermore, this had a positive impact on their emotional well-being, which led to increased confidence and a reduction in anxiety.

#### 3.5.4. Implementation

Generally, participants were happy with how the implementation of the programme ensued (Table 7). Some also reported that their feelings of isolation were reduced through engaging with the programme. One key factor was the “friendly” faces of the staff who delivered the programme. This potentially highlights the need for suitably qualified staff to deliver an exercise programme.

#### 3.5.5. Maintenance

Participants reported that more social spaces for patients with PCC would have enhanced their experience in the programme (Table 8). The program was run so that the majority of time was spent on a one-on-one basis between trainers and participants. There were also some concerns raised about the post-program maintenance of exercise given that, once the program was complete, the participants were not offered any further support for their rehabilitation. This presents some potential for future research to implement related changes in order to maximise this programme’s suitability for PCC patients.

### 3.6. Focus Group at 12 Months

Participants expressed feelings of embarrassment about their illness and that difficulty explaining often invisible symptoms led to social withdrawal and frustration. Moreover, they emphasised their desire to improve but felt unheard and dismissed by those who questioned the reality of their condition. These findings are consistent with those reported by Owen, Ashton, and Skipper et al. [36]. These elements paint a picture of individuals struggling with a debilitating condition that affects not just their health, but also their ability to connect with others, fostering a sense of isolation and frustration aligned with findings reported by Gerlis et al. [35]. Participants discussed the disruption of their usual roles, leading to a negative impact on their family lives. The inability to fulfil their responsibilities, coupled with increased frustration and exhaustion, created a sense of disarray, and impacted everyone’s well-being. The described sense of disarray and sense of being a burden on partners resonates with the findings of Gerlis et al. [35] and Daynes et al. [8], emphasising the broader social and familial consequences of PCC. Furthermore, the frustration stemming from the inability to fulfil responsibilities and comparison with milder cases echoes the experiences shared by participants in Gerlis et al. [35], who expressed surprise at the severity and duration of their symptoms. The loss of independence and normality, with basic tasks becoming impossible, led to an increased reliance on their partners and other family members. This shift in the marital dynamic, coupled with the feeling of being a burden, suggests a potential strain on the relationship. These experiences showcase the far-reaching effects of PCC. The illness not only impacts the individual’s health, but also their social life, mental well-being, and family dynamics. However, as participants’ symptoms gradually improved, they displayed an increase in their confidence by week 6. An important component is that at the 12-month time point, the positive effects of the PCCRRP were still being felt. Generally, participants were “back to normal” at 12 months and cited both physical and mental health rewards (Table 9). This is unique of the current study and suggests that the physiological changes reported in this study via quantitative methodology assisted with these long-term benefits.

For PCCRRP improvement, participants emphasised the importance of self-management tools, including information packets with symptom management strategies and prescribed exercise exit plans for home or local gyms. Establishing routines for exercise, healthy eating, and self-care activities were also highlighted as crucial for sustained recovery. Furthermore, the FG also revealed valuable insights for programme optimisation. Participants expressed a desire for a supportive network fostered through social media groups, meetups, or group exercise sessions with multiple instructors. This aligns with the findings of Gerlis et al. [35], who emphasised the value of shared experiences and peer support during rehabilitation, highlighting the importance of social connection in the recovery process. Moreover, a hybrid model with both group and individual exercise sessions was suggested for increased flexibility. Additionally, participants emphasised the need for advocacy with employers to facilitate time off work for exercise, recognising its importance for recovery.

#### 3.6.1. Reach

Five participants (n = 3 female, 2 male) took part in the PCCRRP 12-month FG.

By incorporating a wider range of support for cognitive difficulties, the PCCRRP can empower patients to regain control of their daily lives and improve their overall well-being. However, it is important to acknowledge that the needs of PCC patients are diverse, and the pilot programme will require adjustments to address a wider range of challenges. Previous studies [9,10,11] acknowledge the diverse needs of patients recovering from PCC and emphasise the importance of tailoring rehabilitation programmes to individual needs and abilities. Additionally, as highlighted by multiple participants who felt the programme would have been more helpful “a lot sooner” after their diagnosis, ensuring early intervention through increased programme availability and outreach efforts is crucial. This aligns with previous evidence which emphasise the benefits of starting rehabilitation early after hospital discharge or the onset of LC symptoms [11]. The positive experiences shared by participants in Table 10 highlight the potential for such programmes to make a significant impact. Twelve months later, all participants continually expressed a “light at the end of the tunnel” in respect of their recovery. They also provided emphasis on proactive strategies like going out for walks and connecting with others highlight the potential of such programmes to make a significant impact.

#### 3.6.2. Effectiveness

Table 10 displays improvements in various aspects of physical health and mental health, including reduced anxiety, depression, and fear, and increased confidence and social interaction after participating in the programme. Participants expressed how utilising physical activity and improving overall fitness helped contribute to a more positive mental outlook. Tangible gains in physical fitness and strength helped to foster a sense of improved confidence and autonomy that had a positive improvement on their QoL. These findings align with previous studies [8,9,10,11,35] which also reported similar improvements following exercise-based rehabilitation programmes, except the current study also offers context through the focus group design. Some participants experienced improved cognitive function and reduced brain fog after the programme. While some maintained physical activity after the programme, others struggled with motivation and consistency. Indeed, some experienced challenges in maintaining a fitness routine, particularly when faced with external disruptions. This highlights the importance of finding a routine that works and the negative consequences when it is not adhered too. Furthermore, the FG discussions revealed a compelling need for more comprehensive PCCRRPs. Beyond physical limitations, participants described significant mental and emotional challenges, highlighting the need for a holistic approach. One participant powerfully described their desperation due to a lack of available support and information, emphasising the urgency of making resources more accessible (Table 11). This pilot programme might be effective for some, but its impact could be significantly amplified by ensuring wider availability and outreach efforts to reach more patients in need. Generally, the FG discussions underscored the importance of hope, empowerment, access to information, and social connection, similar to findings previously reported Rutherford et al. [14] for PLWPCC. While the programme has demonstrably improved the lives of participants, there is room for improvement to ensure it reaches a wider population and offers a more holistic approach that addresses the multifaceted challenges of PCC.

#### 3.6.3. Maintenance

The 12-month FG displays the multifaceted challenges of PCC. The FG yielded a rich tapestry of experiences, highlighting the complex and far-reaching impacts of the illness. While the rehabilitation pilot programme might be effective in some areas, the participants’ narratives suggest a need for a more comprehensive approach. Table 12 portrays findings which underscore the importance of acknowledging the diverse needs of PCC patients. Rehabilitation programmes can be more effective by incorporating support for mental health, cognitive difficulties, daily routine management, and overcoming logistical challenges. By equipping patients with a wider range of tools and strategies, these programmes can empower them to manage their symptoms, improve their overall well-being, and adjust to their new normal (Table 12).

### 3.7. Summary of Findings

A community-based ERS intervention, the PCCRRP, effectively improved physical function and QoL in individuals with PCC. The programme, delivered by an FCCT, was evaluated using the RE-AIM framework to provide comprehensive insights into its reach, effectiveness, adoption, implementation, and maintenance. The key findings were (1) Rapid and sustained improvement: The PCCRRP demonstrated significant positive impacts within the initial six weeks, with large effect sizes observed across most outcome measures. For those adhering to the programme (adopters), these improvements in PA levels, fatigue, QoL, and confidence in daily living were sustained over a 12-month period. (2) Holistic benefits: Beyond physical improvements, participants reported significant positive changes in their social lives and mental well-being. (3) Addressing barriers to participation: The programme successfully alleviated anxieties and fears associated with PA, primarily due to the safe and supportive environment fostered by staff and peers who understood the participants’ condition. (4) Shared experience: An unexpected finding was that participants highly valued the opportunity to share their experiences with others in a similar situation during the FGs, highlighting the importance of social support in recovery from PCC.

These findings align with previous research in a similar FCCT setting by Rutherford et al. [14], reinforcing the effectiveness of community-based interventions for individuals with chronic health conditions. Moreover, the present study’s mixed-methods evaluation may effectively support FCCT report outcomes to justify future funding. Indeed, the present study further contributes to a growing body of evidence on the potential benefits of exercise and rehabilitation for individuals with PCC, demonstrating exercise was well tolerated by participants with no reports of incidence of PEM, ME/CFS, or PESE as previously reported in the literature [9,10,11]. Markedly, although the present study demonstrated significant positive impacts within the initial six weeks, a key feature of the study is the longitudinal design which provides valuable insights into the sustainability of the intervention’s benefits over a chronic timeframe. In addition, the current study is uniquely placed in that it blended both quantitative and qualitative outcomes to the rehabilitation programme, which is not the case across the PCC literature currently. Consistent with our results, Wright et al. [12] found that LC is linked to decreased PA and loss of independence. However, contrary to our findings, Wright et al. [12] reported that most participants experienced a worsening of LC symptoms with PA. In contrast, the present study found no exacerbation of PCC symptoms, which is likely due to the personalised rehabilitation approach.

Gerlis et al. [35] reported improvements in physical symptoms (breathlessness and fatigue) and emotional well-being (confidence and reduced worry) through a rehabilitation programme. The current study is unique in that it can be inferred from the findings that physical improvements (quantitative variables) were intertwined with the psychological improvements (qualitative focus group feedback) reported from the participants. For example, a significant improvement in respiratory muscle function was reported by participants, and this reflected the gradual return to physical activity and exercise of the participants reported in the focus groups. Furthermore, the importance of the safe and supportive environment provided by the PCCRRP was noted. Moreover, the staff’s empathy and the camaraderie among peers were highly valued. The programme offered validation and assurance for individuals experiencing ongoing SARS-CoV-2 symptoms, which were often misunderstood or misbelieved by others. This finding aligns with more recent research [35,36] reporting participants feel unheard, dismissed, or poorly informed, with reports of medical gaslighting and inadequate support received. The opportunity to share experiences and reflections with others in a similar situation was considered invaluable. The positive impact on physical and mental well-being reported by participants is consistent with the current study.

To optimise the long-term benefits of programmes like the PCCRRP, future research should consider several key areas. In terms of self-management, comprehensive guides are provided at the programme’s start covering topics such as fatigue, breathlessness, sleep, cognitive issues, exercise routines, healthy eating, mindfulness, and cognitive training. Research should analyse the effectiveness of these specific strategies in improving long-term outcomes. A daily routine, encourages regular physical activity, self-care, and the use of reminders/note-taking tools; it is recommended that researchers should investigate the impact of adherence to these routines on long-term symptom management and well-being. Regarding social support, which facilitates connections through online groups or physical meetups, research should evaluate the effectiveness of different models, such as online versus in-person, on long-term outcomes. Whilst the current study provides support through an individualised approach to rehabilitation, it is important for future research to consider how this can be scaled up to be applicable to more people whilst still capturing quantitative and qualitative outcomes. We acknowledge that this method of rehabilitation is likely costly, so alternatives would be attractive. Nonetheless, it is important to establish if more cost-effective methods lead to similar improvements, and they would only be worth it if so. The proposed research recommendations can be categorised based on their cost-effectiveness. Highly cost-effective options leverage existing programme data or resources, such as analysing the programme’s initial phase or the effectiveness of self-management strategies. Moderately cost-effective research may involve additional data analysis or potentially require small-scale studies to explore areas like the impact of programme structure or daily routine adherence. Research with lower cost-effectiveness signifies a greater need for additional resources. This could involve extending the programme duration, implementing more intensive interventions, or developing highly personalised programme adjustments, all of which may require increased staff time or programme modifications.

### 3.8. Strengths and Limitations

A key strength of this study is its mixed-methods approach to evaluating a community-based intervention, which provides in-depth participant accounts to enhance understanding of outcome data. Furthermore, the study’s longitudinal design, with data collected at baseline, 6 weeks, 12 weeks, 6 months, and 12 months, provides valuable insights into the sustainability of the intervention’s benefits over time. This allowed the researchers to assess the long-term effectiveness and maintenance of the PCCRRP on participants’ physical (fatigue) and psychological function. Moreover, utilising the RE-AIM framework throughout the evaluation process allowed for a comprehensive investigation of the intervention’s effectiveness and implementation, elucidating the factors that contributed to its success and areas that could be improved. Additionally, we were able to demonstrate the efficacy of a community-based intervention for improving the physical and mental function of people with PCC. This research may also provide commissioners and policymakers with detailed information to support future funding applications and resource allocation. A key aspect to highlight is the high frequency of support that was provided from exercise science and physiology staff. These staff were trained in most cases to an undergraduate or post-graduate level, which may necessitate the need for similarly qualified staff in other studies of programs such as this one.

Some limitations must be acknowledged for the current study. We recognise that the instruments used in the studies were not specific to PCC and, instead, are validated against the general population. Whilst not specifically validated in PCC, the instruments still offer value, and in the absence of PCC-specific questionnaires, are still applicable in the study. We also acknowledge the need for further investigation with a larger sample size. A larger sample size improves the generalisability of the findings and statistical procedures (i.e., increase statistical power), making them more applicable to a wider population. This study served as a pilot investigation to explore PLWPCC. The homogeneity of our sample, predominantly comprising white British participants, limits the generalisability of our findings to more diverse populations. Additionally, we acknowledge the absence of a control group. Therefore, we cannot directly attribute all of the observed benefits to the PCCRRP or understand the influence of factors such as support from previous rehabilitation specialists or social interaction with other participants.

## 4. Conclusions

The present study utilised a mixed-methods methodology to investigate the effects of an exercise rehabilitation and recovery programme on the physical function and QoL of individuals with PCC. The results of the present study demonstrate that the PCCRRP was effective in providing PLWPCC with a positive impact on their physical (fatigue) and psychological function, with some returning to levels they previously experienced before their PCC diagnosis. This is a unique finding given that positive outcomes were still being felt at the 12-month focus group. This pilot study offers in-depth recommendations for future studies, where the scalability of this programme could be an attractive option for fellow FCCTs or healthcare settings. It is worth noting that programmes such as the current study will not be the most cost-effective option but do provide significant positive outcomes that may warrant higher prices. Furthermore, these findings suggest that fostering a supportive community and adapting programmes to individual needs are key elements for long-term success in PCC rehabilitation.

## Figures and Tables

**Figure 1 ijerph-22-01672-f001:**
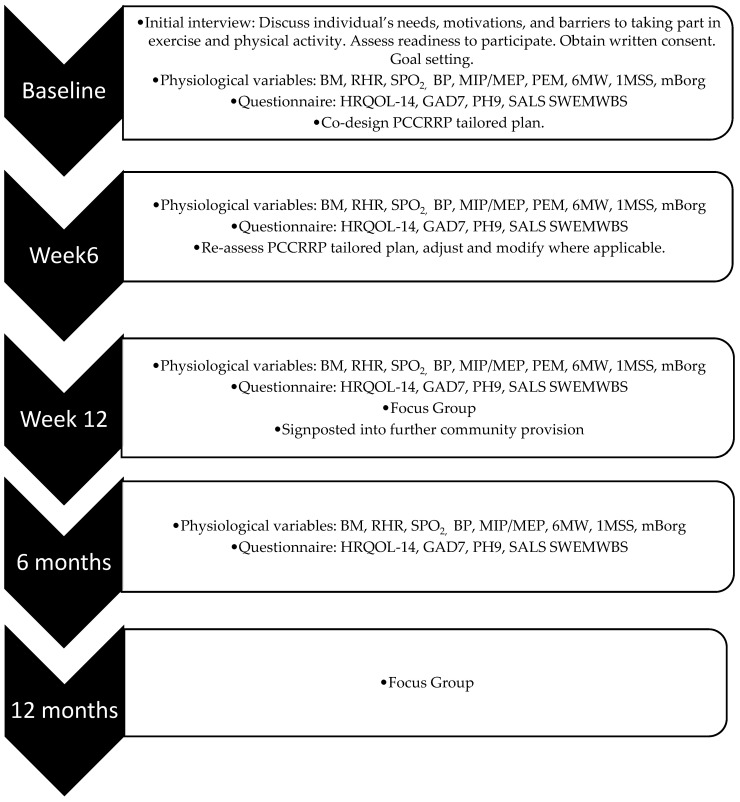
Timeline displaying the outcome measures of PLWPCC in the PCCRRP. Key: Body mass (BM). Resting heart rate (RHR). Arterial oxygen concentration (SPO_2_). Blood pressure (BP). Lung function: muscle inspiratory pressure (MIP). Muscle expiratory pressure (MEP). Post-exertional malaise (PEM). Six-minute walk (6MW). One-minute sit to stand (1MSS). Dyspnoea (modified Borg scale 1–10). Health-Related Quality of Life (HRQOL-14). General anxiety disorder (GAD7). Physical Health Questionnaire (PHQ9). Short Active Lives (SALS).

**Table 1 ijerph-22-01672-t001:** Components of the RE-AIM framework in the context of the PCCRRP.

Construct	Application in This Study	Source of Data
Reach	The number, proportion, and representativeness of people living with post-COVID condition (PLWPCC) who participated in a PCCRRP.	Questionnaire data collected at baseline, 6 weeks, 12 weeks, and 6 months. Focus groups were conducted during week 12 and at 12 months.
Effectiveness	The impact of the PCCRRP on physical activity, fatigue, health-related quality of life, and confidence in daily living.	Quantitative physiological data and questionnaire data collected at baseline, 6 weeks, 12 weeks, and 6 months. Focus groups were conducted during week 12 and at 12 months.
Adoption	The profile of PLWPCC who engaged in the PCCRRP includes their physical activity, health status, and reported barriers and facilitators.	Quantitative physiological data and questionnaire data collected at baseline, 6 weeks, 12 weeks, and 6 months. Focus groups were conducted during week 12 and at 12 months.
Implementation	The key PCCRRP intervention design and delivery characteristics that participants reported as being influential in facilitating their adoption.	Focus groups were conducted during week 12 and at 12 months.
Maintenance	The continued engagement of PLWPCC in exercise and physical activity upon exit of the PCCRRP for positive health-related quality of life.	Quantitative physiological data and questionnaire data collected at 6 months. Focus groups were conducted during week 12 and at 12 months.

**Table 2 ijerph-22-01672-t002:** Summary of demographic profile at baseline. Age is presented as mean ± standard deviation, and other variables as the total number and percentage of total sample.

Variable	Total (%)
**Age**	
52 ± 9 years	
**Gender**	
Male	2 (29)
Female	5 (71)
**Marital Status**	
Married/With Partner	5 (71)
Single/Divorced/Widow	2 (29)
Other	
**Ethnicity**	
White British	7 (100)
**Occupation**	
Care	4 (57)
Education	1 (14)
Manual Labour	1 (14)
Unemployed	1 (14)
**Football Fan**	3 (43)
**Non-Football Fan**	4 (57)
**Fan of Host Club**	0 (0)
**Fan of Another Club**	3 (43)

**Table 3 ijerph-22-01672-t003:** The number (%) of PLWPCC respondents reporting HRQoL-14 activity limitations at baseline, 6 weeks, 12 weeks, and 6 months.

Dimension	Baseline	Week 6	Week 12	6 Months
Are you limited in any way in any activities because of impairment or health problem?	Yes: 7 (100)	Yes: 4 (57.1) No: 3 (42.9)	Yes: 1 (14.3) No: 6 (85.7)	Yes: 1 (14.3) Not sure/Don’t know: 1 (14.3) No:5 (71.4)
What is the major impairment or health problem that limits your activities?	Other impairment/problem: 1 (14.3) Depression/anxiety/emotional problem: 3 (42.9) Lung/breathing problem: 3 (42.9)	Other impairment/problem: 2 (28.6) Lung/breathing problem: 2 (28.6)	Depression/anxiety/emotional problem: 1 (14.3)	Depression/anxiety/emotional problem: 1 (14.3) Lung/breathing problem: 1 (14.3)
For how long have your activities been limited because of your major impairment?	Months: 4 (57.1) Years: 3 (42.9)	Days: 2 (28.6) Months: 1 (14.3) Years: 1 (14.3)	Months: 1 (14.3)	Months: 1 (14.3) Not sure/Don’t know: 1 (14.3)
Because of any impairment or health problem, do you need the help of other people with your personal care needs, e.g., eating, bathing, dressing, or getting around the house?	Yes: 2 (28.6) No: 5 (71.4)	No: 7 (100)	Yes: 1 (14.3) No: 6 (85.7)	No: 7 (100)
Because of any impairment or health problem, do you need the help of other people in handling your routine needs, e.g., household chores, shopping, or getting around for other purposes?	Yes: 5 (71.4) No: 2 (28.6)	No: 7 (100)	Yes: 1 (14.3) No: 6 (85.7)	No: 7 (100)

**Table 4 ijerph-22-01672-t004:** The number of participants (%) reporting HRQoL-14 Healthy Days symptoms, and number of days on which symptoms were experienced, reported at baseline, 6 weeks, 12 weeks, and 6 months by PLWPCC respondents.

Dimension	Baseline	Week 6	Week 12	6 Months
During the past 30 days, for about how many days did pain make it hard for you to do your usual activities, such as self-care, work, or recreation?	5 (71.4) *4 = 30 days* *1 = 23 days* 2 (28.6) *2 = 0 days*	6 (85.7) *6 = 0 days* 1 (14.3) *1 = 3 days*	6 (85.7) *6 = 0 days* 1 (14.3) *1 = 1 day*	6 (85.7) *6 = 0 days* 1 (14.3) *1 = Not sure/Don’t know*
During the past 30 days, for about how many days have you felt sad, blue, or depressed?	4 (57.1) *3 = 30 days* *1 = 20 days* 2 (28.6) *2 = 0 days* 1 (14.3) *1 = Not sure/Don’t know*	3 (42.9) *1 = 10 days* *1 = 5 days* *1 = 2 days* 4 (57.1) *4 = 0 days*	2 (28.6) *1 = 10 days* *1 = 5 days* 5 (71.4) *5 = 0 days*	3 (42.9) *1 = 25 days* *1 = 5 days* *1 = 2 days* 4 (57.1) *4 = 0 days*
During the past 30 days, for about how many days have you felt worried, tense, or anxious?	4 (57.1) *3 = 30 days* *1 = 20 days* 2 (28.6) *2 = 0 days* 1 (14.3) *1 = Not sure/Don’t know*	5 (71.4) *1 = 10 days* *1 = 7 days* *2 = 5 days* *1 = 1 day* 2 (28.6) *2 = 0 days*	3 (42.9) *1 = 20 days* *1 = 5 days* *1 = 3 days* 4 (57.1) *4 = 0 days*	3 (42.9) *1 = 10 days* *1 = 5 days* *1 = 2 days* 4 (57.1) *4 = 0 days*
During the past 30 days, for about how many days have you felt you did not get enough rest or sleep?	6 (85.7) *4 = 30 days* *1 = 15 days* *1 = 2 days* 1 (14.3) *1 = Not sure/Don’t know*	5 (71.4) *2 = 10 days* *2 = 5 days* *1 = 1 day* 2 (28.6) 2 = 0 days	4 (57.1) *2 = 10 days* *2 = 2 days* 3 (42.9) 3 = 0 days	4 (57.1) *1 = 20 days* *1 = 10 days* *1 = 7 days* *1 = 5 days* 3 (42.9) 3 = 0 days
During the past 30 days for about how many days have you felt very healthy and full of energy?	1 (14.3) *1 = 7 days* 5 (71.4) *5 = 0 days* 1 (14.3) *1 = Not sure/Don’t know*	7 (100) *1 = 30 days* *2 = 25 days* *2 = 20 days* *1 = 7 days*	7 (100) *3 = 30 days* *1 = 28 days* *1 = 25 days* *1 = 10 days* *1 = 5 days*	6 (85.7) *3 = 30 days* *1 = 15 days* *1 = 10 days* *1 = 4 days* 1 (14.3) *1 = Not sure/Don’t know*

**Table 5 ijerph-22-01672-t005:** Overall effectiveness themes, sub-themes, and participant quotes related to their experience of the PCCRRP.

Theme	Sub-Theme	Quote
Light at the end of the tunnel	Renewed hope for the future	“Life is back to normal. Back to a bit of a routine. I loved the boxing in the gym, at first, I just couldn’t get the coordination, but the more I did it the better and stronger I got.” (Female)
		“All I wanted was to get back to work and get back to normal. So, coming to the first gym session, I thought great, this is the road back to it. I had become withdrawn and was getting out less and less. These sessions were in my diary, I had to drive to the gym which was great, I got talking with the exercise instructors and other people. That was a big thing for me.” (Male)
		“Feeling normal again. Work, kids, grandkids, shopping.” (Male)
		“The programme gives the belief that there’s hope at the end of this as well, that it’s not just that your stuck and you’re not going to get better. It gives you hope that you can get better, that you can get better and be normal again.” (Male)
		“It’s given me more fight really from being so down and despondent and physically inactive to being able to actually go and do things and plan to do things.” (Female)
Functional Improvements	Physical Function	“I felt my body had let me down and I don’t feel like that now. I feel like it’s stronger and more powerful than it was before. I like seeing people, going for walks with the kids, chasing the dogs and so on, it’s definitely improved.” (Female)
		“We recently went to Disneyland Paris we did 20,000 steps with the kids over three days. I never thought I’d be able to do that. And that was brilliant to be just normal and keep up with them.” (Female)
		“It’s like I have more energy when I go to the gym, than on the days that I have not gone.” (Male)
		“It gives you the motivation. When you’ve gone to the gym and then you come back you can do stuff, whereas if you don’t go to the gym, you will put things off in the day.” (Female)
		“I definitely achieve more when I go to the gym. If I get up and go swimming in the morning, I feel much better.” (Female)
		“Normally I get out of the car and shriek in pain. It just dawned on me the other day, I didn’t, I just got out the car and just walked.” (Male)
Mental Wellbeing	Self-esteem	“My self-esteem improved massively, and taking care of myself, putting on makeup. I mean, I don’t wear a lot of makeup anyway but just making an effort to wear makeup and do my hair and put nice clothes on, rather than slobbing about and not doing my hair.” (Female)
		“Better self-esteem.” (Male)
		“I did not realise that it had improved my self-esteem, but my husband, and my kids would tell me that it had. And I felt more like myself.” (Female)
		“I felt really good. And towards the end of the programme, I started a new job, and we’ve moved house. I hadn’t recognised how far I’d come.” (Female)
	Confidence and Self-Efficacy	“I’m confident. More competent. I’m proud.” (Female)
		“I’m more assertive than I’ve ever been.” (Female)
		“I am back to my happiest self, messing about again and teasing people and I can take it, but I’ll give it as well.” (Female)
		“I do achieve everything. I get the house done, we eat homemade meals and go to the gym. And you know, I don’t know how I manage it when I think about how I was before where I couldn’t do anything.” (Female)
		“Believe in yourself.” (Female)
		“It’s just made me realise, you know, I was really ill and there’s so many more things I want to do and see and achieve. And actually, doing the programme I suppose has given me the confidence I think to kind of do that just kind of push for that.” (Female)
		“So, I’ve just got a promotion at work. So, there was a team leader position in respiratory care, in which I do a lot with the children that are Trachea vented. And I didn’t go for it because I lost my confidence, and I didn’t think I’d be able to manage. So, I didn’t go for it the first time somebody else did, but they didn’t get the job. So, then it was advertised externally and then I applied for it last week and had an interview and I got it.” (Female)
	Advocacy for broader mental health support in the face of the SARS-CoV-2 pandemic.	“I mean, I’ve got to say from a professional point of view, from a mental health perspective, how it helped me with my mental health. It would be nice to see something like this because there are really bad mental issues at the moment. Generally, people coming off the back of COVID, families and everything. So, doing something like this to support mental health alone would be awesome. Really. It would be life changing for not just people who have had COVID, but mental health in general.” (Female)
	Link between physical activity and mental health	“Some people wouldn’t necessarily think physical exercise and how well it just really impacts on your mental health until you’ve actually experienced benefits of it. You know, but people that are generally quite low, if they were referred into a service like this you know, did some things like this exercise, and then made those links, you know it could be life changing for a lot of families as well.” (Female)
		“My mental health was so much better during those 12 weeks on the programme, so when I didn’t go to the gym, please don’t shout at me Steven, but when I was poorly over Christmas, I really felt myself dip! And actually, going back to the gym helped because when I was going in, I felt more resilient about taking things on, and then I stopped going I was like ooh, so I went back! So, I know that is something, my brain goes to me, go to the gym! If I am feeling stressed, go to the gym.” (Female)
		“I felt like that over Christmas, not really got the energy, and don’t really feel like you want to do anything, but you notice yourself feeling down again quite quickly.” (Female)
Exercise rehabilitation and recovery	Motivation	“It’s your positivity, and its tailor made to you and then you give push where it’s needed and just your expertise as well. Like explaining how your body works and how you can make yourself feel better.” (Female)
	Motivation and productivity	“It gives you the motivation. Like say, when you’ve gone to the gym and then you come back home you can do stuff. Whereas if you don’t go to the gym, you will put things off in the day. Like shopping, sometimes when I go to the gym, after I’ll then go to ALDI, to do my shopping, but then, but if I was at home and didn’t do that then I’d be like I really need to go shopping and I would sit there and end up not doing it.” (Female)
		“The exercise instructor was really supportive.” (Male)
		“It’s hard when you go to the gym on your own and you haven’t got someone in your earhole. I would say what do I do now? And then he’s going come on another minute, another minute. And, like, pushing you, push it up another notch.” (Female)
	Tailored exercise sessions	“So, like some things, like I suffer with my back. He would tailor it to help my back if it was playing up, he would put things in you know, to loosen it sort of thing and stretch it out.” (Female)
		“At one point he had me doing circuit training. I was doing all kinds of things, from push-ups, sit-ups, everything. He would have me doing circuits on the cardio machines as well. So, he would have you do 10 min on that, 10 min on that, etc. And I would literally be doing that for an hour. Running between machines! And to be fair he taught me so many new skills in the gym. He knew I would get bored doing cardio for long periods of time. Did my head in! I hated being on anything for longer than 20 min. He knew that I would get bored so he wouldn’t do that.” (Female)

**Table 6 ijerph-22-01672-t006:** Overall adoption themes, sub-themes, and participant quotes related to their experience of PCCRRP.

Theme	Sub-Theme	Quote
Mental well-being	Downward spiral of mental health	“I had no motivation because I had no energy and as a single parent I became really ill with my mental health as well because I thought I was not supporting my children I was letting them down.” (Female)
		“The more I stayed in the more I got withdrawn, especially with working in a school because I felt that if someone sees me outside, they will think he looks alright there’s nothing wrong with him. So, then you start staying in more and more and not going anywhere and not doing anything. And then it’s just a downward spiral.” (Male)
	Ripple effect of PCC on family dynamics	“I just think it affected my whole family because I’m the one that organises things and so because of that nothing else got done. It was just a horrible atmosphere because I wasn’t the one cheering everybody up to do stuff and I was just really grouchy; more grouchy than normal! Just everything was so on top of me, and I just shouted, or I just slept. I would just go and lie down, depressed.” (Female)
		“I had pretty similar to that really and just sort of not being able to do anything and just feeling so anxious and not able to function you know.” (Female)
	Social withdrawal	“I got embarrassed as well because I got to that point where all I did was moan and talk about being ill. Because I’ve been ill so long that I became so embarrassed about it. I stopped talking to my friends because the only thing I had to add to the conversation was moaning about how I was feeling all the time. I was sick of hearing myself saying it! I was sick of explaining how I felt because obviously it’s not something you can see, is it either? So, you know, getting other people to understand what LC was at the time was quite difficult as well because people just think hmm, right, so, you’re making it up. Don’t be silly! You know, it just wasn’t in my head. I definitely felt this way. I just really could not function. And it’s not that I didn’t try because I did. I just couldn’t get a break.” (Female)
	Feelings of injustice and burden	“It was very overwhelming, and it was very frustrating because I saw the people around me get covid and not be as poorly and not have the kind of detrimental day-to-day impact on their lives. They were still able to walk their children up the hill to school, whereas I couldn’t do that. I couldn’t take my children to school; I could barely put clothes in the washing machine and do the housework. And my husband was having to pick up a lot of the slack in the home, it felt more like he was my carer than my husband.” (Female)
	Internal struggle with anxiety	“A big one for me was the anxiety. And there’s just a constant kind of mental battle that I was having with myself daily basis.” (Female)
Physical Activity Adoption	Motivation to Exercise	“So, since I started the programme, I have lost four and a half stone. I have stopped smoking. I have started to do these online challenges. I have completed couch to 5K. I’ve always been able to swim, but I couldn’t swim when I got COVID because I kind of lost the ability to have the rhythm with my breathing. But since joining your programme I’ve swam the equivalent of the English Channel.” (Female).
		“I think it was brilliant! During COVID I struggled to walk my dog and to even step up onto the pavement curb. All the strength in my legs had gone. I got on the cross-trainer and thought, this is my machine. Where I work, we have a gym, so I use it now. I go twice a week.” (Male)
		When we got towards the end of the 12-week programme I felt quite sad because I didn’t want it to end. Since it’s finished, I have joined a gym, and I go with my eldest son. He keeps me accountable, and he is competitive like me which is good.” (Male)
		“The stuff I’ve done, never in a million years is what I would do! Even before I had COVID I never thought I would be a runner. My kids used to joke about what would happen if there was a zombie apocalypse mum? Now I would be like, you can’t catch me! Never did I think I would be able to run ever! I never thought I would lose this much weight.” (Female)
		“The gym has become a hobby; it is no longer a torturous event!” (Female)
		“I’ve spent so many days in the gym, they all know me now! So, before I even start the class, they are getting me doing 20 press ups! I actually found a love for something.” (Female)
Barriers to physical activity	Pre PCCRRP symptoms	“I felt like I’ve got this big wide band across the bottom of my chest and so even breathing normally I struggled, just walking in the house, it was hard and walking up the stairs. I couldn’t take the dogs for a walk because then this band would get tighter and then I couldn’t breathe. And it was just scary that I couldn’t breathe.” (Female)
		“I felt just the same when walking the dog, I would still have to sit down and didn’t have the energy to take my shoes off. Until I joined you guys. I wasn’t going anywhere.” (Male)
	Overcoming pre-existing gym intimidation	“I didn’t have the confidence to use the gym to be honest, but I haven’t got any worries about using it because I’d always been part of it. I would go in to do classes quite happily, but I didn’t really understand what you use in the gym or how to use it or have the confidence to use it.” (Female)
		“I was like yourself because I had that preconceived idea. I’ve always done stuff like that, at school I’d always do stuff, I’d run with the children, I’d play football, I’d do everything like that, and I thought I am keeping myself fit because I’m always nonstop. I’ve got friends who went to the gym, and I thought, no, that’s really not for me. Everyone in there will be like you see on the TV, everyone will look at me and go, no. So going was quite scary at first because it’s like, what am I going to do? How do these machines work? Or I don’t really get it, but after the first couple of sessions like, actually, that’s really good.” (Female)
	Impact on physical ability.	“Well pre-covid I was swimming every day. And I worked with complex children with special needs. So, I’d be lifting, moving, chasing, you know, on the go the whole day. But then as well, I was supporting my family because I’m a single parent and doing all the jobs at home. After covid, I couldn’t swim because I couldn’t breathe properly to swim. And I couldn’t work. There’s no way I could have managed to work and looked after my children.” (Female)
		“I would walk the dog and I noticed when I got in, I would have to sit on the stairs for 5 min before I got the energy to take my shoes off because I just couldn’t do it. It crept up on me, but very quickly, if you like. I noticed if I was by a curb unless I was walking at it, I would have to throw my arms up to get up it. My strength had gone completely out of my legs. I’ve lost my strength.” (Male)
	Limitations in physical and cognitive ability	“Pre COVID-I was pretty much nonstop because I was teaching primary school children and then I’ve got my two boys because I have those on my own. So, I’ve got my two boys and constantly on the go with those. So, one of them plays football, so we’re doing training, we would do football matches. With the little one, we’re out on our bikes. I’d also got another business on the side which was American based. So, I was up till late, and I was making phone calls bringing stuff back and it was using the Car, so you know, moving the trailer around to do lots of trips it was non-stop! And then covid hit and it more or less stopped everything. And then I wasn’t able to stand up, it took the job away because I couldn’t stand up and teach all day. I couldn’t think quick enough, and I tried to go back to work three times on a phased return, and it got to the point I was trying to teach fractions with the year fives, and I just could not think to even work it out.” (Male)
	Loss of confidence and ability to fulfil professional duties	“I’m a scuba dive instructor and I was teaching this lady in the pool just a finning technique, swimming underwater with the fins on, and she was swimming away from me and I thought I haven’t got a hope in hell of catching her up if anything went wrong, and if this was in open water, you know it would be serious. So, I took myself out of the equation, I had to stop instructing. Thankfully, I am back now since your programme, and I have been signed off for 3 years to instruct again.” (Male)
	Travel	“I found it an issue, because of the distance, having no money, the stress of travelling.” (Female)
		“Bit of a barrier because of my new job. And so financially became a bit of an issue because I was coming from Tamworth.” (Female)
		“You can say the travelling was good because it gave that reflection time on the way home. You’ve got to build yourself up before you got there, but then you could reflect on the way home. So, I did like the travel side of things.” (Male)
	Accessibility	“It was really handy for me; it’s really straightforward five minutes door to door.” (Female)
		“You need a programme in every town and every city! That’s what we need.” (Female)
	Shifting focus from self-care to reintegration into daily life during LC recovery	“As the programme went on, finding time I found was harder, because at the start this was the big thing that I am doing. Whereas, as I started to get better, I was doing more in terms of my family or work.” (Male)
	Desire for social connection and shared experience	“I think it would have been nice to meet everybody at the beginning of the programme. And then we could swap phone numbers and meet at the gym. It would be nice if you didn’t do it every time, but you know, it might not be everybody at the same time, but just a walk around somewhere because we’ve all been through the same thing.” (Female)
		“It would have helped if we had all met earlier on. So, it would have been nice to meet everybody earlier on, just for emotional support, I guess as well. I was on my own. Nobody else had got it, and it would have been so helpful if I knew everyone here and, you know, just talked, I’ve had a rubbish day I’m exhausted, someone to understand.” (Female)
	Social accountability	“So, you could meet and then have a coffee after. Recently I have made excuses not to go to the gym. Whereas I think if I’ve made like plans, or you’d arranged to meet with somebody then you’re more likely and will. And they’ve gone through what you’ve gone through.” (Female)
		“Because, like, when with the four of us did meet up, so halfway through the programme, when we were here for the news, going on TV, I think it was good because it was just sitting, talking very informal. And we talked about it (LC). We did a bit of networking, tying us together, that would have been good then.” (Male)
Motivation for adopting PCCRRP	Feelings of guilt and shame	“After spending five days on the ward on oxygen and eventually coming home with a walking stick. And not being able to do anything, you know, like just literally just being a shadow of my former self really. And it took a good six to nine months to feel better. I was so unhappy, and I never got back to the point that I was before I had covid. There’s a lot of guilt and shame. I think there’s a lot of shame I carried around with me, just like why me? Why did this happen?” (Female)
	Loss of professional identity and confidence	“I have been a teacher for 29 years! It was second nature! I could walk into a classroom and do a lesson off the top of my head and not have to think about anything. I could go and do it. If someone said I can’t do an assembly, can you do it? Yeah. Walk into an assembly, make up stories off the top of my head and…That ability I lost.” (Male)
	Disorienting cognitive effects	“I would be driving somewhere, and the number of times I went, and I’d end up somewhere different. We were going to go to IKEA, and I was going the wrong way up the M6 and it’s like, where am I going to? And I was heading towards Liverpool, not towards Walsall! I don’t know why but I knew I was going somewhere.” (Male)
	Loss of multitasking ability and executive function	“Well, I was like the queen of multitasking before I got covid. I was teaching full time looking after four kids, the house. My husband worked a 60-h week and then just after I got covid I was just overwhelmed I think, constantly with everything. I just couldn’t go into supermarket because I was so anxious about catching covid again and that’s if I could actually get round the supermarket, I couldn’t push the trolley. So, Steve would have to push the trolley. I couldn’t even make a shopping list. It’s just cognitively I was just really struggling to kind of … But before I could be on the phone, planning a lesson, I just couldn’t do any of that anymore at all, it had all gone.” (Female)
	Confidence and coping mechanisms	“Even now I’ve lost my confidence. So, before I had COVID I could go in and teach them off the cuff, but now I’m literally like everything, I want to over plan. And I’m so meticulous about everything because I’m so worried about getting it wrong. I just feel like even though I’ve always been quite an anxious person, I’m able to be outwardly confident, whereas I think that’s kind of gone a little bit. Or gone a lot to be fair. But going to the gym helped with that, it really helped with that.” (Female)
	Yearning for recovery	“Because I’d had spent so long being sat at home and every time I rang the doctor, every time I spoke to someone, like what can you do for me? What’s happening? There was nothing. It’s like what can you give me, is there something to take? And there was no help. It seemed almost all I kept being told was it’ll take time and it’s like well, I can’t just keep sitting at home and wasting time. I want to be doing something. I’m Impatient anyway because I can’t just do nothing. If I broke my leg, they would say it’s six weeks in plaster, then this and then you’re back to normal. And I was like, well, when can I be back to normal? Because all I wanted was to get back to work and get back to normal. So, coming to the first gym session I thought great, this is the road back to it.” (Male)

**Table 7 ijerph-22-01672-t007:** Overall implementation themes, sub-themes, and participant quotes related to their experience of the PCCRRP.

Theme	Sub-Theme	Quote
Staffing	Programme effectiveness	“I would recommend the programme, it works!” (Male)
		“Go and see your GP and get yourself referred to you.” (Female)
	Transformative impact	“Its life changing! it’s changed my life, literally, has changed my life!” (Female)
		“It worked and more because I’m definitely better than what I was before. I am not just where I was before I had COVID I am better than I was before I had COVID.” (Female)
	Hope and encouragement	“It’s not hopeless. You can refer to the programme, and it will really help you.” (Female)
		“It felt like the isolation was ended. Finally going to be able to see people, so, light at the end of the tunnel as a kind of overall theme.” (Male)
	Community and shared experience	“Be aware that there are other people like this as well. You’re not just on your own.” (Male)
	Staff skills, expertise, and attributes	“Your expertise on how your body works and how you can make yourself feel better. It has an educational element to it as well.” (Female)
		“I was scared, but I think it was a friendly, smiley positive person once you arrived.” (Female)
		“They (instructors) believed in you. It wasn’t like you must do this course. But actually, we know you have LC, it was believing you.” (Female)
		“You were put at ease pretty much straightaway.” (Male)
		“It was just nice to have that, just that one person, who knew me. You know, a friendly face.” (Female)
Group setting	Power of shared experience in building social connection and overcoming self-consciousness	“If we had met each other before, you’d build that natural support, wouldn’t you? You know how we were all in exactly the same position. So, you know, you wouldn’t feel kind of that barrier between me and you, and you’d probably be like, actually, I don’t mind doing the gym session with someone else.” (Female)

**Table 8 ijerph-22-01672-t008:** Overall maintenance themes, sub-themes, and participant quotes related to their experience of a PCCRRP.

Theme	Sub-Theme	Quote
Areas for improvement	Social connection and support	“It would have been nice to meet everybody earlier on, just for emotional support, I guess as well. I was on my own. Nobody else had got it (LC), and it would have been so helpful if I knew everyone here and, you know, just talked, I’ve had a rubbish day I’m exhausted, someone to understand.” (Female)
		“A group session to start, would give you an opportunity to meet people. Chances are we probably would have all started a WhatsApp group then anyway.” (Female)
		“I think it would have been nice to meet everybody at the beginning. And then we could swap numbers and meet at the gym.” (Female)
		“That initial group session would open up more opportunity, and possibly more opportunities to help others more.” (Female)
		“So, we could go for a walk around somewhere and then have a coffee after. I think if I’ve made plans, or you’d arranged to meet with somebody then you’re more likely to and will. And they’ve gone through what you’ve gone through.” (Female)
		“Because, like, when the four of us did meet up so at the halfway point, when we were here for the TV cameras on Midlands Today, I think it was good because it was just sitting, talking very informal. And we talked about it (LC). We did a bit of networking, tying us together, that would have been good then.” (Male)
Maintenance of activity levels	Motivation and accountability	“I think if I could meet with somebody at the gym I would go, I would go more.” (Female)
Staff communication	Handover of exercise prescription between staff	“The only thing I wanted to mention is that sometimes the contact across different staff members wasn’t ideal. So, it was different guys, each different, but all good! So, everybody’s different. But it was just when I come to have X trainer he massively underestimated where I was.” (Female)
		“I think if X trainer had of looked at my programme a little bit more and we’re on the same level. I think X trainer could have brought in quite a bit more. And I was getting a little bit bored. And where was I was used to having Steve who used to really push me (laughs).” (Female)
Maintaining progress after programme completion	Post-programme exercise plan	“We would like a proper plan of exercise for us to do when we’ve finished the programme. Something we can do in our particular gym.” (Male)

**Table 9 ijerph-22-01672-t009:** Overall effectiveness themes, sub-themes, and participant quotes related to their experience of the PCCRRP at 12 months.

Theme	Sub-Theme	Quote
Improved quality of life	Transformation through progress	“I feel like I’ve really made progress. And I’m not exhausted anymore. I’m going to ski school in March which I never thought I’d be able to do; learn to ski. I’m working three days a week now and taking on more responsibility at work. Problems are not so overwhelming anymore.” (Female)
		“For me it just got my breathing back, and my mental health back to normal.” (Female)
		“At the beginning I didn’t think it would do any good for me. But at the end of it you know, I would say definitely give it a go. It’s like if you have to have counselling. People say oh, you know, I don’t think that will work for me, but I think you have to go and give it a try and see what you get from it.” (Female)
		“I just want to say that even though I wasn’t as bad as the others I found it to be life changing” (Male)
	Physical Improvements	“The programme itself made me feel so much better! All my aches and pains just feel so much better” (Female)
		“I am loads better.” (Male)
	Impact on daily life	“You must remember what life was like before this. It’s like what you said right at the start, is this the new normal. I just feel so much better. But geneally, I feel well, I couldn’t have said that at the start of the programme.” (Female)
	Hope for recovery	“I think it would be interesting. If we hadn’t done the programme, would we still be in that position where we were before? And we would have just settled for thinking this is our life now. Just thinking about it really upsets me, it really does.” (Female)
		“I think it gives you it gives you a lot of hope for the future. For me it did.” (Female)
		“Well, I can remember when I had my own kids, I used to lift them up to ride on my shoulders. So, when I got PCC, we were walking down to the next village where we live, and I was taking the grandkids down. And I remember I struggled to carry them, I had to put them down. But now picking them up again is fine.” (Male)
	Recognised importance of self-care	“Being aware of self-care, being aware in the past I’ve just carried on with my master’s degree and picked up extra days at work and just run myself into a hole. Now I know I need to just spend some time looking after myself.” (Female)
	Assertiveness and self-care	“So, if I don’t want to do something I am not doing it. Normally I would just be a people pleaser, but I don’t need to do it. I have got a lot better at saying no. I think that has come from part of being on this programme.” (Female)
	Increased self-esteem and self-efficacy	“I have been able to publicly speak which is something I’ve never been able to do. My most recent talk was at a church a few miles away. I talked about sustainability, and I did some flower arrangements. And again, talking about my COVID journey. I mean, just inspiring really with what I have done, you know, I was very, very ill and the fact that I’m doing this, that’s a positive for me, you know.” (Female)
	Improved mental health	“I just thought I don’t have time for it, I don’t want the drama. I just really want to enjoy my life. I remember, coming back to these meetings makes me think back to that time. I don’t want to go back to that time. When I was incredibly ill, and I thought was going to die! I’m not going to lie I thought was going to die and sometimes I go back to that and then I just don’t want any of this reality and drama. A lot of that comes from the fact that we just use the darker, horrible things we’ve experienced that not really many people understand.” (Female)
		“Exercise is great for that. I think we all must find what we like. I like being outside in the garden and just sitting there in the fresh air.” (Female)
	Shifting priorities	“Being active as become more at the forefront.” (Male)
	Taking ownership of health	“I eat well, drink, all the things you used to advise. And then I feel better.” (Female)
Exercise to support mental health	Mind–body-connection	“Mentally I know that I have to do some exercise now. If I don’t do any exercise my brain struggles. I have to do something; I have to go to the gym. I need to go for a walk, I need to do something. Otherwise, I can spiral downwards and make myself feel quite low.” (Female)
Self- management	Relapse	“Once we were on the programme, my sleep improved with going to the gym, and doing things because it made me go out and make me do things. That helped massively! It seemed to lapse a bit through the summer. I’m really able to push myself to get back there now. Pick up elements that we did on the programme and try and keep those implemented.” (Male)
Overcoming adversity	Connection and perseverance	“When I was off work for months, I started growing flowers for my well-being and now I do talks to help others about my personal COVID journey, and how it helped me to grow flowers. I think I would have just lost myself because I was so ill. I had the vicar come to see me with the last rights, and he’s a bit of a joker and he said, oh, I’d love to do your funeral. And I think that was the turning point for me. I said you’re not taking me! And it’s the flowers that kept me going. I saw them as my seed bed if you like, to watch them grow. That was stimulating and my focus in life a bit selfish. I’ve got to keep them alive to keep going. And as they grow it just gave me something to keep going.” (Female)
	Maintaining motivation	“And that’s why now, when I do the talks and things you know it’s since this, there will be a light at the end of the tunnel. Don’t give up, believe in yourself, and just keep going. And I think that’s why I push myself hard now, because I don’t want to go back to that dark place.” (Female)
Focus Groups	Social support	“These meetings do help, but I think it would have really helped if I had met everybody I think right at the start. So, we can talk to each other for support as well, so you know you’re not going mad. That would have been really helpful.” (Female)
		“It was an eye opener to see that you had got other people in the gym at the same time who had got the same problems as me.” (Male)

**Table 10 ijerph-22-01672-t010:** Overall adoption themes, sub-themes, and participant quotes related to their experience of the PCCRRP at 12 months.

Theme	Sub-Theme	Quote
Barriers to Gym Attendance	Logistics	“So, it’s kind of a lack of time, and travel. The distance and just the fact that I am not working locally anymore.” (Female)
	Inconvenient location	“I did sign up to the gym, but because I was working part time and going to the gym in Burton, and I live in Stafford. I would go on my lunch break, or I would try and go here and there, and all through the summer. But since I have dropped down to part time at work, I just hardly come to Burton anymore.” (Female)
	Routine	“We are trying to throw everything into our business, and I just have no routine.” (Male)
Alternative Physical Activities	Importance of enjoyment	“I don’t go to the gym anymore, but I think it’s because I get really bored on my own. It’s good if you have someone there to push you on. But I’m just walking the dogs regularly, doing lots with the kids, taking the kids on bike rides, you know, getting on with things to try to build into my day-to-day life.” (Female)
		“So, we started going out on our bikes with little one. I also park further away from our shop so, I had to do a 10 min walk every day.” (Male)
Self-Care Prioritisation Challenges	Change in priorities	“I think I have gone back to how I was pre-COVID that everyone else is my first priority. I need to make sure everyone else is okay first, before me.” (Female)
Value of Social Support	Spending time with friends	“It’s like a support away from your family and all that. People who haven’t got LC. It lifts your spirit. Makes you feel good, having a laugh.” (Male)
	Home-based exercise	“So, perhaps producing some sort of guide maybe, with stuff that you could do at home, exercises at home.” (Male)
Impact of Programme Timing	Referral pathway	“I just wish that I could have got on the programme a lot sooner. I think it would have helped a lot sooner and I think I would have felt a lot better sooner because it was two years before I got on the programme.” (Female)

**Table 11 ijerph-22-01672-t011:** Overall implementation themes, sub-themes, and participant quotes related to their experience of the PCCRRP at 12 months.

Prioritising and maintaining self-care	Loss of external support and accountability	“On the programme you would have the instructors keep reminding us we need to prioritise our own health first. But now, I’ve got no one. No one I’m accountable to as such, I am accountable for myself.” (Male)

**Table 12 ijerph-22-01672-t012:** Overall maintenance themes, sub-themes, and participant quotes related to their experience of the PCCRRP at 12 months.

Theme	Sub-Theme	Quote
Routine and goal setting	Maintaining social interaction	“You know, having PCC thinking is this what life is going to be like now. I’m not going to get back to normal. The programme helps get back to normal. And the fact of going out twice a week helped me massively because it was written down. I’ve got to be there at nine o’clock on a Tuesday morning. I knew that it was almost like a non-negotiable to me that, so I had to do it and I’m almost having to set myself my negotiables now that I am finished on the programme. I must talk to three different people. I have to set these soft targets like that to make sure I do.” (Male)
		“It just makes you feel valued. That if I weren’t around people would miss you and it just lifts you, even more. It’s great for your own self-worth.” (Female)
Prioritising and maintaining self-care	Internal conflict between self-care and family obligations	“With everything that’s going on, and again, this is just one thing, I’ve started to put myself a little bit down the list of priorities in terms of where I fit. Because I’ve got to make sure my parents are alright, my boys are alright.” (Male)
	Boundaries and self-care	“Putting boundaries in place and saying actually that if I just keep working, I am no good to anybody. So, you need to have a break and a means of tracking to put things away, so that you dedicate time to having fun with friends and family.” (Female)
The societal pressure to prioritise others over self-care	Internal conflict between self-care and societal expectations	“It’s all about you, it’s all about yourself and we want to be seen to be pleasing people. At home I look after my partner and our animals. So, I think we’re all creatures where we always put ourselves last in terms of priorities. Then we’ll just make time to go for a haircut or grab a coffee every now and whenever you can. So, it’s not your priority, is it? If it is your priority, if it would seem to be a priority, I think people are quick to say oh, she’s having her nails done, she’s having her hair done blah, blah, blah. But really, we should all be doing that as well. We should all be able to do our own thing because I’m not okay.” (Female)
Mental and emotional impact of post-COVID condition	The challenge of managing panic attacks and anxiety	“When you are that ill, all you are concentrating on is trying to get through that day. If you’re having a panic attack, you’re just sitting there praying, like literally praying that you’re going to get through the next hour.” (Female)
	Cognitive difficulties	
		“I never used to carry anything. And now I’ve got notebooks I’ve got post- it notes in this pocket, there’s also a pen in this pocket. So, the most compulsive behaviour that I’ve gotten is to have all these things with me, because otherwise I won’t remember it.” (Male)
		“I asked the doctor about it, and he said do you know when you’re doing it? And I said yes. And he said good, because it’s when you don’t know you’re doing it you need to worry.” (Female)
		“Mine is up and down at the moment. It’s like with the brain fog, I have had that come back again. My manager always says to me why have you got a paper diary. I need it for my calendar, I always fill that in. I have to write everything down.” (Female)
		“I think when you’ve got brain fog as well, sometimes it’s hard to think about routine because you can’t think about what to do first. Even the simplest task. I have struggled this week, I have got out of bed, got dressed, made my bed, and then I think should I brush my teeth now or later. My brains going like, what do I do first”? (Female)
		“That’s what pretty much finished me off when I was teaching fractions. I was at the blackboard, and I couldn’t remember, and my mind just went blank. I mean even now with my youngest son, he’s only eight years old, and we play memory games or do jigsaws and I just struggle. I got in the car to drive to Ikea, and I was going completely the wrong way on the M6 in the opposite direction.” (Male)
		“It’s more when I get stressed or flustered and then it goes totally. I mean working in the shop is helping by making me talk to people and get that back, but I’ll pick up some material or I can get a price for something and then I will get to the till, and I will have forgotten what that price was. But then I feel silly and stupid and ridiculous. Because you can’t do that.” (Male)
		“I kind of laugh it off. I’m quite happy myself, anything that happens like that. I just brush it off. It doesn’t bother me.” (Male)
	Fear of missing out and overexertion	“I feel like I don’t want to stop and think so I just keep on pushing on. I have blocked in loads of things to do with the kids that we would never normally do, because I just do not want to stop. So just keep going. And it’s like, one of the things that’s happened with me since having LC and being in hospital is that I do not want to say no to opportunities, because I’ve got a lot of doorways open. There’s lots of opportunities. And when you think you’re not going to have any opportunities, it’s nice to have opportunities. But that’s when the self-care boundaries kick in and actually, let’s just make sure I am okay.” (Female)
		“I have had a lot of mood swings going up and down a lot recently, because I couldn’t think of that, I used to be on my knees praying before I go to bed a night. And I think that’s what I’ve had over the recent few weeks again. I might have had COVID again and not known it.” (Female)
Importance of healthy eating	Adopting healthy eating habits	“Since the programme finished; I have realised I need to eat properly. More vegetables eat more healthfully. So that has been more of a challenge. And that has been a focus for us, when we do eat properly, we feel better for it.” (Male)

## Data Availability

The data presented in this study are available on request from the corresponding author due to legal restrictions.

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
