# Peer review of "A Mixed-Methods Evaluation of a Post-COVID-Condition Rehabilitation and Recovery Intervention Delivered in a Football Club Community Trust"

_ijerph, 2025, doi:10.3390/ijerph22111672_

Round 1
Reviewer 1 Report
Comments and Suggestions for Authors
Thank you for the opportunity to review this small pilot study exploring the impact of a post-Covid-Condition Rehabilitation intervention delivered in a football club community trust setting.
This is described as a mixed method study using the RE-AIM framework to assess aspects of the programme over 12 months. It used a prospective design for collecting quantitative data including physiological assessments and questionnaires, and two focus groups. While this is a mixed methods study, there is no clear data integration of both quantitative and qualitative data, but it appears to be an explanatory sequential design where quantitative data on reach, effectiveness, adoption and outcomes maintenance are explained with the use of focus groups to explore programme implementation.
The background is established, although there are more recent meta-analyses on exercise for long covid that could have been referred to.
Programme and setting is clearly described, although the exercise programme content was not detailed apart from ‘supervised low to moderate intensity’ with a ‘combination of aerobic, stability, mobility and strength-based’ exercises. The programme was delivered over 12 weeks, two times each week, with plans co-developed by participants and exercise leaders. Duration of each exercise session is not provided, but physiological parameters such as resting heart rate, and SPO2 were measured prior to exercise.
Participants were referred to the programme after assessment at a local long-covid clinic, and were considered to require support with rehabilitation. Clear inclusion criteria are provided, however the clinical reasoning behind how ‘requiring support with rehabilitation’ was established is not described. Potential participants with post-exertional malaise or chronic fatigue were excluded.
The study only reports on those completing all measures and there is no description of the proportion of participants who did not complete. It is not possible therefore to determine some of the ‘adoption’ elements of the RE-AIM framework. While analysing only completers aligns with some other studies of this type, it does limit interpretation.
The study is underpowered to detect effects, and while the authors use ANOVA to determine changes over time, conclusions can’t be drawn from such small samples. An alternative approach to that used in this study could be to report on minimal clinically important differences, reporting for each individual rather than using statistical analyses. Ultimately, statistically significant differences matter less than clinically important differences for participants, particularly in a sample size this small.
The focus groups are partially described, but the questions used were not provided. Furthermore, the lead author is described as having ‘cultivated rapport…throughout the 12-week intervention period’. The role of the lead researcher is not disclosed, and apart from the statement regarding rapport, there is no reflexivity statement. Two problems arise here: the relationship between the author and participants may influence responses in the focus groups, with demand characteristics eliciting positive experiences; while without reflexivity statements it is not possible to determine the assumptions and perspectives held by those involved in both data collection and analysis.
The authors describe following Braun and Clarke (2006) but don’t appear aware of the more recent iterations of Braun and Clarke’s approach to thematic analysis (eg Braun, V., & Clarke, V. (2020). One size fits all? What counts as quality practice in (reflexive) thematic analysis? Qualitative Research in Psychology, 1-25. https://doi.org/10.1080/14780887.2020.1769238). I also note that ‘data saturation’ is described, suggesting that the authors have not read Braun and Clarke’s consideration of saturation in thematic analysis (see Braun, V., & Clarke, V. (2020). One size fits all? What counts as quality practice in (reflexive) thematic analysis? Qualitative Research in Psychology, 1-25. https://doi.org/10.1080/14780887.2020.1769238).
Thematic analyses are largely descriptive rather than analytic, and I felt a missed opportunity to really understand the meaning of this programme on participants. A greater depth of analysis might help explore Reach – why participants were identified and what attracted them to the programme (why football club?); Effectiveness – what effectiveness meant to these participants, how did they know it had been effective; Adoption – what participants saw as the ‘main ingredients’ in their rehabilitation, particularly in light of comments about the social element being an important component; Implementation – what were barriers and facilitators to engagement, why did they continue to participate over time; and Maintenance – what elements of the programme were retained, what participants saw as crucial to maintaining the gains they’d made.
While the study has considerable limitations in terms of sample size, probably unhelpful statistical analysis, and a fairly superficial qualitative analysis that did not integrate both forms of data, it did use a helpful framework for analysing knowledge translation, and the authors did gather a considerable amount of data. A mixed method methodology provides for a nuanced examination of such programmes, and I felt the authors could have delved a little more deeply into the individual trajectories and qualitative data to provide readers with a more satisfying analysis with practical implications.
Author Response
Reviewer one
Reviewer comments
Thank you for the opportunity to review this small pilot study exploring the impact of a post-Covid-Condition Rehabilitation intervention delivered in a football club community trust setting.
This is described as a mixed method study using the RE-AIM framework to assess aspects of the programme over 12 months. It used a prospective design for collecting quantitative data including physiological assessments and questionnaires, and two focus groups. While this is a mixed methods study, there is no clear data integration of both quantitative and qualitative data, but it appears to be an explanatory sequential design where quantitative data on reach, effectiveness, adoption and outcomes maintenance are explained with the use of focus groups to explore programme implementation.
The background is established, although there are more recent meta-analyses on exercise for long covid that could have been referred to.
Added to line 69: A systematic review and meta-analysis of 23 studies including 1579 individuals displayed positive effects on PCC related symptoms including fatigue, dyspnoea and depression, furthermore improvements in overall QoL
Zheng, C., Chen, X., Sit, C. H. P., Chen, L., Zhang, Y., & Chen, T. (2024). Effect of Physical Exercise-Based Rehabilitation on Long COVID: A Systematic Review and Meta-analysis. Medicine and Science in Sports and Exercise, 56(1), 143–154.
Programme and setting is clearly described, although the exercise programme content was not detailed apart from ‘supervised low to moderate intensity’ with a ‘combination of aerobic, stability, mobility and strength-based’ exercises. The programme was delivered over 12 weeks, two times each week, with plans co-developed by participants and exercise leaders. Duration of each exercise session is not provided, but physiological parameters such as resting heart rate, and SPO2 were measured prior to exercise.
Participants were referred to the programme after assessment at a local long-covid clinic, and were considered to require support with rehabilitation. Clear inclusion criteria are provided, however the clinical reasoning behind how ‘requiring support with rehabilitation’ was established is not described. Potential participants with post-exertional malaise or chronic fatigue were excluded.
Added paragraph to line 120: Following the PCC clinic assessment the need for rehabilitation is established when symptoms are complex in nature and causing significant functional impairment. Thus, requiring a supervised and structured multidisciplinary approach to recovery.
The study only reports on those completing all measures and there is no description of the proportion of participants who did not complete. It is not possible therefore to determine some of the ‘adoption’ elements of the RE-AIM framework. While analysing only completers aligns with some other studies of this type, it does limit interpretation.
In accordance with Moreton et al. [35] and Rutherford et al. [15] a "completers only" approach was adopted to analyse the outcome data. Completers were defined as individuals who provided usable questionnaire data at baseline, 6 weeks, 12 weeks, and 6 months. This approach aligns with that of Moreton et al. [35] and Rutherford et al. [15] which provides the most effective method for determining changes in physical activity and HRQOL over time, considering the constraints of the data collection method. We do note that we have 100% completion throughout the study measures of those who initially took part.
The study is underpowered to detect effects, and while the authors use ANOVA to determine changes over time, conclusions can’t be drawn from such small samples. An alternative approach to that used in this study could be to report on minimal clinically important differences, reporting for each individual rather than using statistical analyses. Ultimately, statistically significant differences matter less than clinically important differences for participants, particularly in a sample size this small.
In response, we have added more information to the limitations to recognise the lack of statistical power. This is why we included the effect size, which is a useful way to determine changes outside of relying on p value derived interpretation. We therefore feel this is sufficient, but we are happy to discuss further.
The focus groups are partially described, but the questions used were not provided.
The following are the 12-week focus group questions:
- Describe all your symptoms when you were diagnosed with long COVID.
- How did you function with daily activities before you got COVID-19, and how was it afterwards?
- How useful was the patient information provided to you at the time of your diagnosis?
- How did you feel when you were referred to our PCCRRP?
- How did you feel about the PCCRRP being delivered by a FCCT?
- How did you find traveling to the PCCRRP? Was it a positive or a barrier?
- Since you completed our PCCRRP, how were your symptoms after? What did you feel like after the 12 weeks?
- How do you feel mentally and physically on completion of the program, and how has it impacted your daily life?
- Did you feel supported by our staff on the PCCRRP?
- Would you recommend the PCCRRP to other people?
- Are there any suggestions for improvements for the PCCRRP?
We would be happy to provide this as supplementary material post publication
Furthermore, the lead author is described as having ‘cultivated rapport…throughout the 12-week intervention period’. The role of the lead researcher is not disclosed, and apart from the statement regarding rapport, there is no reflexivity statement. Two problems arise here: the relationship between the author and participants may influence responses in the focus groups, with demand characteristics eliciting positive experiences; while without reflexivity statements it is not possible to determine the assumptions and perspectives held by those involved in both data collection and analysis.
The authors describe following Braun and Clarke (2006) but don’t appear aware of the more recent iterations of Braun and Clarke’s approach to thematic analysis (eg Braun, V., & Clarke, V. (2020). One size fits all? What counts as quality practice in (reflexive) thematic analysis? Qualitative Research in Psychology, 1-25. https://doi.org/10.1080/14780887.2020.1769238). I also note that ‘data saturation’ is described, suggesting that the authors have not read Braun and Clarke’s consideration of saturation in thematic analysis (see Braun, V., & Clarke, V. (2020). One size fits all? What counts as quality practice in (reflexive) thematic analysis? Qualitative Research in Psychology, 1-25. https://doi.org/10.1080/14780887.2020.1769238).
Corrected and most up to date APA 7th Edition
Braun, V., & Clarke, V. (2021). One size fits all? What counts as quality practice in (reflexive) thematic analysis? Qualitative Research in Psychology, 18(3), 328–352. https://doi.org/10.1080/14780887.2020.1769238
Revised text:
Each FG consisted of a sex-mixed composition. Seven participants attended the 12 weeks FG, while five attended the 12-month FG. The FGs were both held at Burton Albion Football Community Centre lasting approximately 90 minutes and were digitally recorded. Guided by the principles of reflexivity the thematic analysis targeted trustworthiness and authenticity of findings [40]. The lead author (SR) and Co-authors (LG & IK) served as note-takers, recording key themes and sub-themes emerging from the FG discussions. After each session, these themes and sub-themes were verbally presented back to the participants for confirmation and clarification, ensuring the accuracy and authenticity of the captured information. This participant-led verification process ensured the validity and precision of the acquired information. Thus, demonstrating participant experience and privilege of voice over the researchers own perspectives was a key part of the design of the research process. Such deliberate strategy enhances Vigor and validity of the thematic analysis [40].
The lead researcher (SR) had cultivated rapport with the participants throughout the 12-week intervention period, fostering a sense of familiarity and trust, which was vital for facilitating open and honest dialogue. This pre-existing connection was established during the baseline consultation and PCCRRP delivery, as well as during recruitment and participation in the FGs. Similarly, existing relationships among participants further enhanced engagement and interaction during the discussions. Additionally, it highlights the positive impact of these connections on participant engagement and interaction and consequently helps generate meaningful results.
FGs were administered at week 12 and 12 months respectively. Following the FGs, the recorded audio files underwent verbatim transcription by the lead author (SR) and co-author (SL). Guided by the principles of reflexivity [40] the thematic analysis designed to elicit authenticity of findings. Thematic analysis utilised the process of familiarisation, coding, generating themes, reviewing themes, defining and naming themes, and writing up [40]. For the interview data, after transcription and immersion in the transcripts, coding revealed intriguing features within the data. These features were subsequently grouped into coherent themes using the RE-AIM framework components. A visual map was hand-crafted to illustrate the themes and their interrelationships. Four researchers’ (LG, IK, AH & AK) met to refine the specifics of the themes and to generate clear definitions and names for them, which were then shared with the lead author (SR). This approach has commonly been used in the investigation of football-led health improvement Programs [15, 33, 41]. A collaborative approach to data collection was utilised to mitigate any potential for researcher bias.
Thematic analyses are largely descriptive rather than analytic, and I felt a missed opportunity to really understand the meaning of this programme on participants. A greater depth of analysis might help explore Reach – why participants were identified and what attracted them to the programme (why football club?); Effectiveness – what effectiveness meant to these participants, how did they know it had been effective; Adoption – what participants saw as the ‘main ingredients’ in their rehabilitation, particularly in light of comments about the social element being an important component; Implementation – what were barriers and facilitators to engagement, why did they continue to participate over time; and Maintenance – what elements of the programme were retained, what participants saw as crucial to maintaining the gains they’d made.
Line 2 page 14: Reach
Seven participants took part in the PCCRRP (n=5 female, n=2 male). Participants expressed being drawn to the FCCT setting due to it feeling less clinical than a traditional GP surgery. However, others stated their desperation to improve their health, and well-being was as such that they were willing to try anything to get the help and support they felt they needed.
Line 12 page 14: Effectiveness A recurrent theme identified throughout the FG discussions was the concept of a "light at the end of the tunnel," signifying a sense of hope. Additionally, “feeling normal”, regaining a sense of pre-Illness functioning. There was a strong emphasis on the participants' desire to return to their usual activities and capabilities. In some instances, beyond their original capabilities’ pre-diagnosis
Line 27 page 17: Main ingredients to rehabilitation: Participants reported a significant factor in their improvement was the PCCRRP focusing on personalised exercise plans. Participants expressed the tailored exercise plans empowered them to manage their symptoms and helped them to regain a sense of control over their lives. Furthermore, this had a positive impact on their emotional well-being which led to increased confidence and a reduction in anxiety.
Page 20. Table 7:
|
Travel |
“I found it an issue, because of the distance, having no money, the stress of travelling”. (Female) |
|
|
“Bit of a barrier because of my new job. And so financially became a bit of an issue because I was coming from Tamworth”. (Female) |
|
|
“You can say the travelling was good because it gave that reflection time on the way home. You've got to build yourself up before you got there, but then you could reflect on the way home. So, I did like the travel side of things”. (Male) |
|
Accessibility |
“It was really handy for me; it's really straightforward five minutes door to door”. (Female) |
|
|
“You need a program in every town and every city! That’s what we need” (Female) |
While the study has considerable limitations in terms of sample size, probably unhelpful statistical analysis, and a fairly superficial qualitative analysis that did not integrate both forms of data, it did use a helpful framework for analysing knowledge translation, and the authors did gather a considerable amount of data. A mixed method methodology provides for a nuanced examination of such programmes, and I felt the authors could have delved a little more deeply into the individual trajectories and qualitative data to provide readers with a more satisfying analysis with practical implications.
We do agree to a certain extent that this and any paper could have a more in-depth analysis of the either the quantitative or qualitative elements. One must consider the word count, length and readability of the manuscript, which is already substantially large. Our opinion is that the depth of both quant and qual could be improved, but not in the format of a mixed-methods investigation. Hence, why this may come across this way. Of course, happy to discuss further.

Reviewer 2 Report
Comments and Suggestions for Authors
Dear Authors,
Great study - thanks for the opportunity to review it. Your findings are very important, and extremely interesting. Overall, this is an extremely well presented, detailed and informative study. Well done to all involved. I have a few questions, some are based on my own curiosity, however. My one area of concern (and concern is not really the most appropriate word to use) is the magnitude of data - there's a lot and reading table after table after table dilutes the significance somewhat. But I appreciate that much data was collected given it's mixed-methods, but I do wonder if a figure or a diagram could be used to help navigate what was done and what was said.
Below are my comments.
Line 52: COVID-19 is mentioned previously yet here SARS-CoV-2 is referenced. I’d suggest sticking with one term throughout (possibly SARS-CoV-2 given that this is official title).
Line 70: Strength training or resistance training? There is a difference as training for strength implies something different compared to general strength and conditioning and resistance training.
Line 96: Perhaps add a line that explains what a community trust is. There will likely be a few readers who do not make the necessary link.
Line 107: Is a reference needed for the RE-AIM framework?
Line 133: Am assuming measurements were recorded immediately prior and post exercise?
Line 153: Who facilitated the focus groups?
Line 280: You’ve included CI (confidence intervals) in the results yet there is no mention of these in your statistical data reduction and analysis. If these are to remain, I would suggest including in the appropriate section (at 95% one would assume).
Results
General comment: Your results are very interesting, particularly in that BM and blood pressure did not change throughout, yet other physiological variables did. I appreciate that you measured a lot here, and there is a lot to take in and consider. I had to read the results numerous times to fully comprehend what was done and the changes, or not, involved. The use of acronyms doesn’t help, but I appreciate the necessity. I did wonder if there is a better way of displaying some of the results? Also, given that participants attended the gym, was relative measures of strength (via a 1RM or similar) taken? I’m curious more than anything as resistance training is known to improve BM and can help manage BP in certain situations.
Reach and effectiveness
Line 18-19: I may have missed this, but how you formalised your themes and sub-themes are unclear. I note that overall method you used, but how did you, as authors, finalise the codes and themes? Essentially, the actual process is missing.
The quotations listed in Table 6 are informative and telling, yet when participants ‘life is back to normal’ (theme: light at the end of the tunnel), were any quotations obtained that detailed what life was life prior to the study?
The results are very well presented, but I return to my initial point in that it is unclear how themes are sub-themes were developed. There is a theme Advocacy for broader mental health support in the face of the COVID-19 pandemic, but how did this emerge? What other voices supported this theme?
Page 24:
Line 33: I recall one participant mentioning that they were embarrassed, under the theme social withdrawal, were there others? Was there a point in time or point in the program that such feelings subsided in the participants? Was this linked to confidence, perhaps?
Page 25-26
Line 87. You state the Table 10 highlights improvements in various aspects of mental health, but some of the themes listed in Table 10 don’t necessarily reflect a mental health construct. For instance, Improved quality of life appears to be more of a physiological reaction rather than a mental health one. If improvements to fitness were gained this would likely shift mental health to a more positive outlook, giving confidence and autonomy. Perhaps allude to this a little more as the themes of physical activity and overall improvements to health are much more apparent than their improvements to mental health.
Page 31
Summary of findings
The purported value of staff involved in the program has clearly resonated with many participants. Aside from subject matter knowledge, did the staff engage in positive psychological assistance to participants? Were coping skill, self-talk skills used by staff to help participants?
Line 139: A supportive environment is mentioned here. I’d like to know how staff provided such a supportive environment.
Author Response
Reviewer two
Dear Authors,
Great study - thanks for the opportunity to review it. Your findings are very important, and extremely interesting. Overall, this is an extremely well presented, detailed and informative study. Well done to all involved. I have a few questions, some are based on my own curiosity, however. My one area of concern (and concern is not really the most appropriate word to use) is the magnitude of data - there's a lot and reading table after table after table dilutes the significance somewhat. But I appreciate that much data was collected given it's mixed-methods, but I do wonder if a figure or a diagram could be used to help navigate what was done and what was said.
Response: We did think about this for some time in terms of how we could present what is a sea of data across quantitative and qualitative elements. We would be against changing it at this stage as our belief is that readers will pick up the sections, they need within the paper, rather than tackle it in one go (if that makes sense). We also think the depth helps with “why” some things work (or don’t), which we feel is extremely lacking with long COVID exercise rehab studies thus far. Many just do a short 4-8 week intervention with only quantitative outcomes and with no follow up, which we feel is poor science. We also acknowledge the table approach can seem repetitive, but we feel using a framework such as RE-AIM is really important to dissect the data and help the reader focus on particular important elements of this type of research.
Below are my comments.
Line 52: COVID-19 is mentioned previously yet here SARS-CoV-2 is referenced. I’d suggest sticking with one term throughout (possibly SARS-CoV-2 given that this is official title).
23 replacements made to SARS-CoV-2
Line 70: Strength training or resistance training? There is a difference as training for strength implies something different compared to general strength and conditioning and resistance training.
Changed to resistance training.
Line 96: Perhaps add a line that explains what a community trust is. There will likely be a few readers who do not make the necessary link.
Added the sentence FCCT’s are the charitable arm of a football club.
Line 107: Is a reference needed for the RE-AIM framework?
Reference added
Line 133: Am assuming measurements were recorded immediately prior and post exercise?
The sentence already states Additionally, resting heart rate (RHR), blood pressure (BP), and arterial oxygen concentration levels (SPO2) were measured and recorded prior to exercise.
Line 153: Who facilitated the focus groups?
Sentence added “The lead author (SR) and Co-authors (LG & IK) facilitated the FG. This information is included in more detail from line 212”
Line 280: You’ve included CI (confidence intervals) in the results yet there is no mention of these in your statistical data reduction and analysis. If these are to remain, I would suggest including in the appropriate section (at 95% one would assume).
Added to line 255: Now correct, sorry for missing this. Statistical data are presented with 95% confidence intervals to indicate the precision of the estimated values.
Results
General comment: Your results are very interesting, particularly in that BM and blood pressure did not change throughout, yet other physiological variables did. I appreciate that you measured a lot here, and there is a lot to take in and consider. I had to read the results numerous times to fully comprehend what was done and the changes, or not, involved. The use of acronyms doesn’t help, but I appreciate the necessity. I did wonder if there is a better way of displaying some of the results? Also, given that participants attended the gym, was relative measures of strength (via a 1RM or similar) taken? I’m curious more than anything as resistance training is known to improve BM and can help manage BP in certain situations.
Improvements were observed in BP and BM but just not of statistical significance. Table 7 shows a quote from one participant’s weight loss which adds nuance to the numerical findings we observed.
We did not take 1RM measures as maximal type testing is not recommended in this population and we would face backlash for this. We did not relate the exercise they were undertaking to any threshold unfortunately, as we felt the freedom to tailor their exercise to their own boundaries was important (such as using the pacing concept). Anecdotally, Given the varied nature of their exercise patterns it wouldn’t strictly relate to a threshold or allow us to recommend a threshold of exercise.
Reach and effectiveness
Line 18-19: I may have missed this, but how you formalised your themes and sub-themes are unclear. I note that overall method you used, but how did you, as authors, finalise the codes and themes? Essentially, the actual process is missing.
We have added the below to describe this process:
Refer to line 260: Thematic analysis utilised the six-step process of familiarisation, coding, generating themes, reviewing themes, defining and naming themes, and writing up [40]. For the interview data, after transcription and immersion in the transcripts until saturation, coding revealed intriguing features within the data. These features were subsequently grouped into coherent themes using the RE-AIM framework components. A visual map was hand-crafted to illustrate the themes and their interrelationships. Four researchers’ (LG, IK, AH & AK) met to refine the specifics of the themes and to generate clear definitions and names for them, which were then shared with the lead author (SR). This approach has commonly been used in the investigation of football-led health improvement Programs [15, 33, 41].
The quotations listed in Table 6 are informative and telling, yet when participants ‘life is back to normal’ (theme: light at the end of the tunnel), were any quotations obtained that detailed what life was life prior to the study?
Some more information on this can be seen in Table 7 displays multiple quotes relating to physical and mental health of what life was like prior. Table 6 didn’t lend itself to this discussion in any detail.
The results are very well presented, but I return to my initial point in that it is unclear how themes are sub-themes were developed. There is a theme Advocacy for broader mental health support in the face of the COVID-19 pandemic, but how did this emerge? What other voices supported this theme?
This was part of the thematic process as highlighted in answer two comments above. This is also now in the manuscript.
Page 24:
Line 33: I recall one participant mentioning that they were embarrassed, under the theme social withdrawal, were there others? Was there a point in time or point in the program that such feelings subsided in the participants? Was this linked to confidence, perhaps?
Added to page 25 line 54: However, as participants symptoms gradually improved, they displayed an increase in their confidence by week 6, so much so that some participants requested early cessation of the program at this mid-way point due to feeling stronger physically and mentally. Feelings of embarrassment diminished and they displayed improved confidence. Participants were reminded that positive change had been a result of their commitment to the PCCRRP and that these feelings may continue to improve as they progress to week 12 and beyond. This links into the quotes for overall effectiveness in table 10
Page 25-26.
Line 87. You state the Table 10 highlights improvements in various aspects of mental health, but some of the themes listed in Table 10 don’t necessarily reflect a mental health construct. For instance, Improved quality of life appears to be more of a physiological reaction rather than a mental health one. If improvements to fitness were gained this would likely shift mental health to a more positive outlook, giving confidence and autonomy. Perhaps allude to this a little more as the themes of physical activity and overall improvements to health are much more apparent than their improvements to mental health.
I think some of this would depend upon what quality of life is. For example, in the WHO QoL questionnaire mental health constructs are apparent, and this is the interpretation of QoL from the authorship team. We do acknowledge the point though, and our example below I guess speaks to your comment here.
Page 26. Line 101: Table 10 displays improvements in various aspects of physical health and mental health, including reduced anxiety, depression, and fear, and increased confidence and social interaction after participating in the program. Participants expressed how utilising physical activity and improving overall fitness helped contribute to a more positive mental outlook. Tangible gains in physical fitness and strength helped to foster a sense of improved confidence and autonomy that had a positive improvement on their QoL.
Table 10:
|
Exercise to support mental health |
Mind-body-connection |
“Mentally I know that I have to do some exercise now. If I don't do any exercise my brain struggles. I have to do something; I have to go to the gym. I need to go for a walk, I need to do something. Otherwise, I can spiral downwards and make myself feel quite low”. (Female) |
Page 31
Summary of findings
The purported value of staff involved in the program has clearly resonated with many participants. Aside from subject matter knowledge, did the staff engage in positive psychological assistance to participants? Were coping skill, self-talk skills used by staff to help participants?
Thank you for your valuable feedback and for raising this important question regarding the psychological support provided during the rehabilitation program.
The protocol was strictly designed to deliver physical activity and fitness interventions, and our staff were primarily trained in exercise physiology and rehabilitation. Therefore, our staff's role was confined to the delivery and supervision of the physical training component, and they were not tasked with providing explicit psychological assistance such as coping skills or self-talk skills.
While we did not formally incorporate these psychological support components into the study's design, we acknowledge the significant overlap between physical recovery and mental well-being. As exercise was supervised, an organic but informal approach to these psychological aspects would have occurred. As our results suggest, the improvement in physical capacity appeared to have a positive secondary effect on participants' mental health. We agree that this is a critical area for future research.
We believe that future studies could greatly benefit from a multi-disciplinary approach, integrating both physical rehabilitation and structured psychological interventions to explore their combined impact. We have included a point on this in our revised discussion section to reflect the importance of this perspective. We have added a couple of sentences into the strengths and limitations section suggesting the level of staff training could have been important.
Line 139: A supportive environment is mentioned here. I’d like to know how staff provided such a supportive environment.
Page 23:
|
Staff skills, expertise and attributes |
“Your expertise on how your body works and how you can make yourself feel better. It has an educational element to it as well”. (Female) |
|
|
“I was scared, but I think it was a friendly, smiley positive person once you arrived”. (Female) |
|
|
“They (instructors) believed in you. It wasn’t like you must do this course. But actually, we know you have LC, it was believing you”. (Female) |
|
|
“You were put at ease pretty much straightaway”. (Male) |
|
|
“It was just nice to have that, just that one person, who knew me. You know, a friendly face”. (Female) |
The staff provided a supportive environment via encouragement and motivation, empathy and understanding, providing a structured and safe environment and building a sense of community.

Reviewer 3 Report
Comments and Suggestions for Authors
Firstly, I would like to congratulate the authors for undertaking a study on Long COVID, knowing how difficult it is for these patients due to the variability of their symptoms. Likewise, it is essential to help these patients with these sequelae. The study is very interesting but needs some improvements before it can be published. I will now make some recommendations.
1. Authors' affiliations must include the department/university/country. Please follow the author guidelines for affiliations.
2. I recommend that authors review acronyms when they first appear. There are some where the acronym appears first and the text is described afterwards. Make sure that the first time an acronym appears, it is defined. Then, always use the acronym throughout the text. For example, line 130.
3. The RE-AIM framework is very interesting for mixed methods, but I recommend explaining a little about what it consists of and providing a reference.
4. There is a bit of confusion with the tables. Tables should be named in the text and then appear close to and after that text. They should not appear before being named. Table 3 is named before Table 2. This can be confusing for the reader. Please put the tables in order and indicate when they should be consulted. If necessary, change the table number. The reference to Table 3 cannot appear before Table 2 appears.
5. There is no figure 1? Figure 2 is mentioned but there is no figure 1. Please resolve this.
6. The reference for the ‘NHS-approved PX-100 EU Salter fingertip pulse 181 oximeter’ is missing (see lines 181-182).
7. Results section: When opening subsections, an introductory paragraph should be included before a table. Please add a paragraph for the reader's convenience.
8. It is recommended not to create subsections with very short one-sentence paragraphs. It is recommended to combine these subsections into one. See sections 3.1.5; 3.1.6; 3.1.7.
9. Table 5 cannot appear before being named; it must be placed immediately after being named. This provides a more linear reading experience for the reader.
10. It is recommended to include an AI statement as an ethical disclosure, and authors should indicate whether they have used AI and for what purpose. This is a recommendation.
Author Response
Reviewer three
Firstly, I would like to congratulate the authors for undertaking a study on Long COVID, knowing how difficult it is for these patients due to the variability of their symptoms. Likewise, it is essential to help these patients with these sequelae. The study is very interesting but needs some improvements before it can be published. I will now make some recommendations.
- Authors' affiliations must include the department/university/country. Please follow the author guidelines for affiliations.
All complete
- I recommend that authors review acronyms when they first appear. There are some where the acronym appears first and the text is described afterwards. Make sure that the first time an acronym appears, it is defined. Then, always use the acronym throughout the text. For example, line 130.
All acronyms checked
- The RE-AIM framework is very interesting for mixed methods, but I recommend explaining a little about what it consists of and providing a reference.
Line 111: The RE-AIM framework is a model utilised to help plan and evaluate the public health impact of interventions [46]. This study used a mixed-methods retrospective design to employ the RE-AIM framework to evaluate the Reach, Effectiveness, Adoption, Implementation, and Maintenance (RE-AIM) of a Post-COVID-Condition Rehabilitation and Recovery Program (PCCRRP) in a FCCT (Table 1).
Reference added
- There is a bit of confusion with the tables. Tables should be named in the text and then appear close to and after that text. They should not appear before being named. Table 3 is named before Table 2. This can be confusing for the reader. Please put the tables in order and indicate when they should be consulted. If necessary, change the table number. The reference to Table 3 cannot appear before Table 2 appears.
Corrections made
- There is no figure 1? Figure 2 is mentioned but there is no figure 1. Please resolve this.
Corrected
- The reference for the ‘NHS-approved PX-100 EU Salter fingertip pulse 181 oximeter’ is missing (see lines 181-182).
Reference added
- Results section: When opening subsections, an introductory paragraph should be included before a table. Please add a paragraph for the reader's convenience.
Added: Results are presented detailing the quantitative and qualitative results separately. The discussion then integrates these findings presenting interpretation in a combined manner providing a comprehensive understanding of the research.
- It is recommended not to create subsections with very short one-sentence paragraphs. It is recommended to combine these subsections into one. See sections 3.1.5; 3.1.6; 3.1.7.
This has now been simplified into 3 sub-headings instead of 9.
- Table 5 cannot appear before being named; it must be placed immediately after being named. This provides a more linear reading experience for the reader.
Corrected
- It is recommended to include an AI statement as an ethical disclosure, and authors should indicate whether they have used AI and for what purpose. This is a recommendation.
Now added

Reviewer 4 Report
Comments and Suggestions for Authors
The article presents an innovative and relevant study, with a sound methodological structure and clear social relevance. However, the main limitations relate to the small sample size, insufficient integration of mixed methods, and the need for a more critical discussion of the results. Overall, it constitutes an interesting contribution, but revisions are required to strengthen methodological clarity, statistical consistency, and critical interpretation.

Author Response
Reviewer four
The article presents an innovative and relevant study, with a sound methodological structure and clear social relevance. However, the main limitations relate to the small sample size, insufficient integration of mixed methods, and the need for a more critical discussion of the results. Overall, it constitutes an interesting contribution, but revisions are required to strengthen methodological clarity, statistical consistency, and critical interpretation.
We appreciate the reviewer's thoughtful feedback and have revised the manuscript accordingly. All edits have been highlighted in yellow. Revisions were made to address the identified limitations concerning sample size, the integration of mixed methods, and the critical discussion of results. We believe the updated manuscript more effectively conveys the study's contribution and meets the reviewer's expectations for methodological clarity and critical interpretation.
The article presents an original study evaluating a community post-COVID-19 rehabilitation programme (PCCRRP) implemented within a Football Club Community Trust. It employs a mixedmethods design (quantitative and qualitative) framed by the RE-AIM model, incorporating physiological measures, questionnaires, and focus groups. The findings indicate significant improvements in participants’ physical function, quality of life, and mental health, although the sample size is very small (n=7). The study provides preliminary evidence that community-based programmes delivered by local sports organisations may be effective in rehabilitating individuals with PCC. It also presents longitudinal data (up to 12 months), which adds value compared with previous studies with shorter follow-up periods. Furthermore, it integrates analyses of physical, psychological, and social impacts, emphasising the relevance of community-based approaches for public health. Strengths: Application of the RE-AIM theoretical framework, providing a comprehensive evaluation across reach, effectiveness, adoption, implementation, and maintenance. A mixed-methods design: robust quantitative measures (6MWT, 1MSS, spirometry, anxiety/depression scales, HRQoL-14) complemented by qualitative analysis through focus groups. Highlights the importance of community support, emphasising the role of social and sporting networks in patients’ well-being
We thank the reviewer for their comments.
Article The topic is current, relevant, and pertinent to public health, given the impact of PCC. The article is well-structured and clearly written, but could benefit from: - A methodological review addressing the aspects mentioned below, including consistency in the presentation of abbreviations (e.g., PEM and ME/CFS appear before being defined)
We have now addressed the abbreviations throughout the manuscript, so they are now accurate.
statistical measures. - Improvements in the presentation and discussion of results, with greater clarification on how the qualitative findings complement the quantitative data (integration of methods)
We have now added more discussion on the interrelationships between the quant and qual data. Please see lines 173-175 179 – 185, lines 208-211, and PG24, lines 65-70.
Limitations: Extremely small sample size (n=7), which limits generalisability and reduces statistical robustness. Quantitative analysis: although p-values and effect sizes are reported, the description of statistical robustness could be more detailed (e.g., full confidence intervals, discussion of potential type II error).
We have added further discussion of the small sample size and we fully acknowledge this within the study. Lines 240-243
Mixed-methods integration: while both quantitative and qualitative results are presented, it is not clearly evident how the two datasets interact or complement each other. Instrument validation: it would be pertinent to indicate whether the instruments used (e.g., HRQoL-14, GAD-7, PHQ-9) have been validated for the PCC population within the UK context.
We have provided more discussion on the interplay of qual and quant outcomes, where the exact places are two comments above. We acknowledge the instruments used are not in the PCC population, so we have added this as a limitation to the study. In lines 246-249 we have highlighted this limitation.
Selection bias: recruitment via a specialised clinic and voluntary participation may limit representativeness.
We acknowledge this and have highlighted this as a limitation
Discussion: could further explore comparisons with existing literature and provide a more indepth consideration of methodological limitations.
We hope with our changes we have no addressed this
Review The authors present the problem and the aims of the study in a contextualised way.
Title: The title is clear, informative, and makes the focus of the study clear.
Abstract: The abstract is comprehensive and well-structured, presenting the objective, methods, results, and conclusions. However, it could more explicitly introduce the topic and highlight the main limitation (small sample size). It is recommended that the IMRaD structure be adopted, with the objective clearly stated in the introduction.
We have opted to keep our abstract in the same format as it currently appears. We have, however, added a sentence to highlight the small sample size.
The use of abbreviations in the abstract should be avoided (e.g., FCCT, QoL).
Now changed
Keywords: It is recommended to replace “Post-COVID-Condition” with the MeSH term “Post-Acute COVID19 Syndrome”: https://meshb.nlm.nih.gov/record/ui?ui=D000094024
We have used the post-COVID-condition term throughout the manuscript so we will keep this the same throughout.
Exercise rehabilitation with “exercise therapy”: https://meshb.nlm.nih.gov/record/ui?ui=D005081 Intervention with “treatment outcome”: https://meshb.nlm.nih.gov/record/ui?ui=D016896 The term “football club community trust” can be retained as a keyword. Although it is not a MeSH term, it is appropriate for contextualising the study and enhancing discoverability, particularly in relation to the specific community-based intervention described.
We have now made those amendments
In addition, although “mixed methods” is not an official MeSH term, its inclusion as a keyword could be considered, as it accurately reflects the study’s methodological approach and may improve discoverability for readers interested in research combining quantitative and qualitative methods.
We have now included this
Note on Introduction: The introduction is well-founded, citing relevant literature to contextualise the topic of PCC and the existing knowledge gaps, thereby justifying the relevance of the study. The relationship between the context—football club community trusts (FCCTs)—and the identified gap in the literature could be more clearly articulated at the end of the introduction.
We have now added a two of sentences at the end of the introduction to highlight this more directly
Lines 132–135: Were the physiological parameters evaluated used as guides for the intervention and the programming of the applied programme’s intensity? At any point, could they have indicated changes that warranted suspension of the programme? This issue should be addressed both in this section and in the subsequent section on exclusion criteria.
They were not used to inform the intervention/exercise intensity, as this was designed with more holistic factors in mind.
We did use a questionnaire to determine the suitability to exercise: “The DePaul Post-Exertional Malaise Questionnaire (DPEMQ) was administered before each exercise session to assess post-exertional malaise (PEM) and determine exercise suitability [17,18].”
We did have tolerances for each variable but that would be too much description for the manuscript.
Lines 121–125 – Intervention Context and Setting: Was any respiratory rehabilitation included in the intervention? For instance, in the introduction, inspiratory muscle training exercises are highlighted as important complementary components of the rehabilitation programme. Additionally, the results include respiratory function data, which should be more clearly articulated.
Now included in lines 199-200
Line 175: Should this refer to Figure 1?
Now corrected
Line 187 – DPEMQ: Are this and other instruments validated for the study population (e.g., GAD7, PHQ-9, SWEMWBS, HRQoL-14)?
They are not validated specifically in long COVID, but have been used in studies with other health conditions. There is a lack of specific long COVID instruments that we could have used, especially considering the study was conducted in 2021/22.
Lines 205–207: “FG was crafted to align with the RE-AIM framework's components. This approach ensured that the discussions focused on the programme's reach, effectiveness, adoption, implementation, and maintenance.” Upon reviewing Table 1, the focus group is only referenced under the “implementation” and “maintenance” phases.
Thank you for spotting this oversight. This has now been corrected.
Lines 200–227: It would be helpful to indicate whether specific software was used for qualitative analysis (e.g., NVivo, Atlas.ti).
No software was used for this analysis
Lines 233–236 (Data Reduction and Analysis): The authors adopted a “completers-only” approach for analysis. This contrasts with the stated concern for statistical robustness, as the sample size is already small and exclusion increases potential bias. While not a writing error, the methodological implications should be discussed more thoroughly. The choice of a completersonly approach requires greater justification, as it may introduce bias. The methodological structure is robust, yet the statistical section could benefit from further detail (e.g., justification for ANOVA with such a small sample).
We have now added that all participants who enrolled completed all parts of the study, so we did not exclude anyone (providing they met the inclusion criteria of course).
We recognise that the ANOVA has limitations, but we used this for the following reasons, 1) it is widely used so can offer a comparison to other literature, and 2) is stronger than repeated t test which can increase error. We also made the decision to include the effect size and confidence interval to compliment our statistical approach. With this in mind, we feel we have provided a comprehensive analysis that is above and beyond most papers in the literature. We have also acknowledged the small sample size in the discussion.
Lines 239–242: The Shapiro–Wilk test is correctly applied; however, with such a small sample size (n = 7), it would be expected that the sample distribution is non-normal. With this in mind, would it not be preferable to use non-parametric tests for variables that appear to be normally distributed?
We found that the variable this test applied to was normally distributed.
Line 267 – Figure 2: Should this be Figure 1? The caption, “Figure 2. Timeline displaying the outcome measures of PLWPCC on the PCCRRP,” appears after the figure, whereas it should precede it according to MDPI style guidelines (the journal in which IJERPH is indexed). The figure legend uses numerous unexplained abbreviations (e.g., “1MSS,” “mBorg”), making it difficult to interpret for readers unfamiliar with them. Visually, it resembles a chronological table rather than a figure and could be presented as a table with footnotes explaining new abbreviations. If retained as a figure, the source should be indicated (i.e., the authors themselves). Finally, the strategy for integrating quantitative and qualitative data should be explicitly described. How was this relationship established?
We have now tweaked this figure to make it clear to the reader. We have added a small section in lines 288-292 to describe how the two types of analysis complimented each other and our justification for this approach.
Note on the results: The results are well organised and report both statistical significance and effect size. The integration of qualitative and quantitative findings could be made clearer. Sections 3.1.1 to 3.2.2: When reporting p-values and stating that significant differences were observed, it is important to also specify the statistical tests that were used. This provides transparency, ensures reproducibility, and allows readers to appropriately interpret the results, particularly since different tests have distinct assumptions (e.g., parametric vs. non-parametric, paired vs. independent comparisons).
We are not familiar with reporting the specific test for each p value reported, and this would not be the norm within the discipline. We have used the statistical tests as per described in the statistical analysis section unless otherwise stated (including ES, and CI).
Some analyses (e.g., GAD-7, PHQ-9) show improvements, but without significant differences between time points; this should be addressed and discussed.
Now amended
Line 275: A brief introduction could be provided prior to the presentation of Table 2.
This has now been added
Lines 279–286: The paragraph “This improvement was significant and substantial at all time points” should be clarified, especially in contrast with “with the exception of week 12 vs. 6 months.”
Excellent point – now corrected
Lines 304–308: Improvements over time in MEP are described as significant (p = 0.024), but it is immediately followed by the statement that “no significant differences were observed at later time points.” This may confuse the reader regarding whether meaningful differences actually occurred.
We are unsure how this can be worded in another way as this is the statistical outcome.
Lines 323–329 (SALS): The text mentions a “significant increase in SALS scores,” yet several pairwise comparisons are not significant (p > 0.05). This oscillation between asserting significance and then qualifying it creates interpretative inconsistency.
We agree – and this is similar to the comment above. We are unsure how we can amend this as the outcomes of statistics is more of a commentary so leaves us with no way to provide interpretation until the discussion.
Table 5 is not referenced in the text.
Now amended
Page 14: Regarding the focus groups (FG), the methodological strategy for analysing their content should be explicitly stated. Tables 7, 8, and 9, as well as Tables 11 and 12, should be properly presented and referenced within the text. Following the presentation of the tables, the discussion should be improved, integrating both quantitative and qualitative data and interrelating them more effectively.
We have made a number of revisions (see earlier comment) that now mean this comment is correct. All tables are now cited in text. We have also provided a small discussion above tables 7,8, and 9.
Note on the conclusion: The conclusion effectively summarises the main findings; however, it could be more emphatic regarding the study's contributions and implications for both practice and academia, particularly within the specific context of its application. It is recommended that the authors explicitly revisit the research objective and demonstrate how it has been achieved, making clear reference to the particular context studied. In addition, might the authors consider highlighting potential avenues for future research? For instance, could the study be replicated in different contexts to examine the generalisability of its findings, or undertaken with a larger sample size to strengthen the reliability of the results? Would it also be valuable to explore related variables or alternative settings that might yield further insights into the topic, should the authors deem this appropriate?
We have now amended the conclusion in an attempt to address this comment. We would be happy to amend further if needed.
References: Adequate diversity of sources is evident, including systematic reviews and meta-analyses. Some references could benefit from the inclusion of DOIs or persistent links to reinforce academic rigour. Several references currently lack a DOI, and some do not follow the proper link format (the “https://” prefix is missing). Incorporating DOIs in references enhances the reliability, accessibility, and credibility of academic work. DOIs provide a persistent identifier for references, making it easier for readers to access the cited material directly, ensuring citation accuracy and consistency, facilitating the tracking of citations and other metrics, and demonstrating adherence to best practices in scholarly communication.
We can address this at proof stage as we were not aware this was in the author guidelines for the journal.
In the reference list, isolated acronyms are generally avoided; the full name of the organisation should be used. For example: National Institute for Health and Care Excellence (reference 1) and National Health Service (reference 45).
Now corrected
The expression “Available online: November 2021” (reference 45) is no longer recommended in formal referencing.
Now corrected

Round 2
Reviewer 1 Report
Comments and Suggestions for Authors
Thank you for the opportunity to review this amended manuscript. It now reads well, and provides a good understanding of the programme and its effects in participants.
Author Response
We thank the reviewers for these comments
Reviewer 3 Report
Comments and Suggestions for Authors
The authors have adequately addressed the suggestions made by both reviewers. The paper is suitable for publication, if approved by the editor.
Author Response
We thank the reviewers for their comments.
Reviewer 4 Report
Comments and Suggestions for Authors
Dear Authors,
Thank you for your thoughtful and comprehensive revisions. The manuscript is now much clearer and stronger, with improved integration of qualitative and quantitative data and a more critical discussion aligned with the study’s objectives.
Two minor suggestions remain:
-
Abstract structure: To better follow the IMRaD format, consider adding a brief introductory sentence that contextualises the problem.
- Keywords: To align with MeSH terminology and improve indexing, keep all the current keywords, but from the four listed below, consider keeping only the two suggested terms:
- Remove “Exercise rehabilitation” keep only “Exercise therapy”;
- Remove “Intervention” keep only “Treatment outcome”.
Best regards
Author Response
Two minor suggestions remain:
-
Abstract structure: To better follow the IMRaD format, consider adding a brief introductory sentence that contextualises the problem.
Author response: we have now amended this in the manuscript
- Keywords: To align with MeSH terminology and improve indexing, keep all the current keywords, but from the four listed below, consider keeping only the two suggested terms:
- Remove “Exercise rehabilitation” keep only “Exercise therapy”;
- Remove “Intervention” keep only “Treatment outcome”.
Author response: we have now amended this in the manuscript